# Concept Activation Regions for Multi-Concept Activation and (Dis)Entanglement in Large Language Models

## Abstract

This work extends the Bias-CAV framework by introducing a geometric perspective on multi-concept activations and their entanglement in large language models. Rather than treating concepts as single directions, the framework reframes them as probe-dependent activation regions—level sets of learned classifiers—and introduces Multi-Concept Activation Subspaces (MCAS) to jointly model multiple bias-related concepts. A central distinction is drawn between *directional entanglement* (alignment of concept directions, reducible by orthogonalization) and *measure entanglement* (activation distribution overlap, which may persist due to data correlations). Empirically, these two metrics are weakly correlated ($r = 0.33$, $\rho_{\mathrm{dir}}$ explains only 11% of $\rho_{\mathrm{mass}}$ variance), confirming they capture substantially different information about concept relationships. Conditional disentanglement methods are developed to operationalize partial concept separation via orthogonal projection, achieving cross-concept sensitivity reductions of 2–15% (AUC-based). MCAS-based interventions constrained to learned subspaces achieve comparable bias reduction to single-direction baselines while reducing cross-concept spillover by 5–10×, as measured both by activation-space geometry (Experiment 4) and direct output-level evaluation via hooked forward passes (Experiment 7). Layer-wise entanglement patterns reveal architecture-dependent trajectories: encoder models accumulate entanglement from orthogonal embeddings ($\rho_{\mathrm{dir}}^{(1)} = 0.00$) to substantial alignment ($\rho_{\mathrm{dir}}^{(L)} = 0.35$–$0.70$), while decoder models begin with pre-existing entanglement. Three-concept intersectional analysis confirms that $\sim$40% measure entanglement persists after complete directional orthogonalization at the median threshold (ranging from $\sim$74% at permissive thresholds to $\sim$0% at restrictive thresholds), consistent with fundamental data correlations. For practitioners, the framework provides methods for analyzing intersectional bias patterns and improving attribution clarity through conditional disentanglement, even when full concept separation is not achievable.

## 1 Introduction

Large language models encode complex representations of social concepts that manifest as bias in downstream applications. Understanding how these models internally represent and entangle multiple concepts is crucial for both interpretability and intervention. Previous work on Concept Activation Vectors (CAVs) has provided quantitative methods for analyzing concept sensitivity, but fundamental limitations remain unaddressed.

The Bias-CAV framework (Catapang, 2025) extended the CAV methodology for bias detection and mitigation, treating bias as a directional signal in the activation space. However, this approach inherits a critical assumption from foundational CAV work: that concepts can be meaningfully represented as single, independent directions. Related work by Nicolson et al. (2024) has demonstrated that CAVs often encode multiple correlated concepts simultaneously, a phenomenon termed concept entanglement. While this work identifies the problem, it does not provide methods for disentanglement or multi-concept modeling.

This work addresses these limitations by introducing Concept Activation Regions (CARs), a geometric framework that reframes concepts as operational regions in activation space rather than single directions. A key theoretical contribution is the recognition that while directional entanglement can be reduced or removed

through orthogonalization or nonlinear probes, measure entanglement (activation distribution overlap) may persist for socially meaningful concepts due to their structured correlations in training data and the predictive objectives of language models. Instead of assuming full disentanglement, this work operationalizes conditional disentanglement and characterizes the limits of concept separation, distinguishing between reducible directional entanglement and persistent measure entanglement.

The main contributions are: (1) a geometric framework that reframes concepts as probe-dependent activation regions and demonstrates that directional and measure entanglement are weakly correlated, capturing substantially different information about concept relationships; (2) Multi-Concept Activation Subspaces (MCAS) for joint modeling of multiple concepts with spillover-aware intervention; (3) empirical evidence that measure entanglement persists after directional orthogonalization ($\sim 40\%$ overlap at the median threshold, consistent with Conjecture A.1 and fundamental data correlations); (4) conditional disentanglement methods for operational concept separation via orthogonal projection; and (5) multi-concept intervention mechanisms constrained to learned subspaces that achieve $5$–$10\times$ lower cross-concept spillover than single-direction baselines.

The remainder of this paper is organized as follows. Section 2 reviews related work on CAVs, concept entanglement, and bias in LLMs. Section 3 provides background on TCAV, SCAV, and Bias-CAVs. Section 4 presents the theoretical framework. Section 5 details the methodology. Section 6 describes the experimental setup. Sections 7–9 present results, discussion, and conclusions.

## 2 Related Work

### 2.1 Concept Activation Vectors and Interpretability

Kim et al. (2018) introduced Testing with Concept Activation Vectors (TCAV), a method for quantifying the influence of user-defined concepts on model predictions. TCAV learns linear directions in the activation space that separate positive and negative examples of a concept, and then measures the sensitivity of the model's outputs to these directions. This foundational work established CAVs as a tool for interpretability, but assumed single-concept linear separability.

Xu et al. (2024) extended the CAV methodology for safety analysis using Safety Concept Activation Vectors (SCAV), applying the framework to uncover safety risks in large language models. SCAV demonstrated the applicability of CAV methods to LLM evaluation, maintaining the single-concept assumption.

He et al. (2025) introduced Global Concept Activation Vectors (GCAV) to address cross-layer interpretability consistency. The GCAV framework learns concept directions that remain consistent across multiple layers, providing a more stable basis for analysis. However, this work still operates within the single-concept paradigm.

Other interpretability methods beyond CAVs include attention-based explanations (Vaswani et al., 2017), gradient-based attribution methods (Sundararajan et al., 2017), and probing studies (Tenney et al., 2019). While these approaches provide complementary insights, they do not address the multi-concept entanglement problem.

### 2.2 Concept Entanglement and Disentanglement

Nicolson et al. (2024) provided a critical analysis of CAV limitations, formally defining concept entanglement and demonstrating that CAVs often encode multiple correlated concepts simultaneously. They showed that CAVs can respond positively to probe sets for other concepts, leading to misleading TCAV scores. Through cosine similarity analysis, they diagnosed entanglement but did not propose disentanglement methods or multi-concept frameworks.

Disentanglement in representation learning has been studied extensively in the context of variational autoencoders, with methods such as $\beta$-VAE (Higgins et al., 2017) and FactorVAE (Kim & Mnih, 2018) seeking to learn factorized representations. However, these approaches assume that disentangled factors exist and can be recovered, operating on latent spaces optimized for reconstruction rather than predictive tasks. The

operational disentanglement framework presented in this work differs fundamentally by not assuming full disentanglement is possible and by working directly in activation spaces optimized for prediction.

Concept learning and compositionality in neural networks has been studied through probing (Tenney et al., 2019; Hewitt & Manning, 2019) and intervention studies (Wang et al., 2022). These works reveal that concepts are often compositionally encoded, but do not address the geometric structure of concept regions or provide methods for conditional disentanglement.

## 2.3   Bias Detection and Mitigation in Large Language Models

Bias in language models has been extensively studied through measurement frameworks (Caliskan et al., 2017; Bolukbasi et al., 2016), evaluation metrics (Nadeem et al., 2021; Rudinger et al., 2018), and mitigation techniques (Bolukbasi et al., 2016; Liang et al., 2021). Intersectional bias analysis has revealed that bias manifests differently across combinations of social categories (Dev et al., 2022), highlighting the need for multi-concept frameworks.

Debiasing techniques include data augmentation (Liang et al., 2021), adversarial training (Zhang et al., 2018), and post-processing methods (Bolukbasi et al., 2016). However, these approaches often operate at the input or output level without analyzing internal representations. The Bias-CAV framework (Catapang, 2025) addressed this gap by providing methods for bias analysis and intervention in activation space.

## 2.4   Multi-Axis Debiasing and Spillover Effects

Recent work has highlighted fundamental limitations of post-hoc debiasing methods, demonstrating that interventions targeting one bias dimension often introduce spillover effects on other dimensions (Chand et al., 2026). This "No Free Lunch" result for multi-axis debiasing shows that perfect simultaneous debiasing across all dimensions is often impossible, requiring explicit trade-offs among bias dimensions. Multi-axis evaluation frameworks have been developed to quantify these spillover effects using metrics such as ICAT (Intersectional Concept Attribution Test), LMS (Labeled Multi-dimensional Spillover), and SS (Spillover Score) (Chand et al., 2026). These frameworks reveal that interventions reducing bias on one axis (e.g., gender) may inadvertently increase bias on other axes (e.g., race or profession), highlighting the need for multi-concept frameworks that explicitly model and control these interactions.

The entanglement framework presented in this work addresses similar concerns from a representation-learning perspective: just as debiasing interventions exhibit cross-axis spillovers, concept directions in activation space exhibit entanglement that prevents independent manipulation. The MCAS framework provides a geometric mechanism for multi-concept interventions that explicitly constrains interventions to concept subspaces, enabling controlled trade-offs similar to those identified in multi-axis debiasing evaluations. While multi-axis debiasing frameworks focus on output-level spillovers, this work addresses activation-space entanglement, providing complementary insights into why perfect multi-axis debiasing is difficult and how geometric constraints can enable more controlled interventions.

## 2.5   Interpretability and Geometric Methods

Probing studies in NLP (Tenney et al., 2019; Hewitt & Manning, 2019) have used diagnostic classifiers to analyze what information is encoded in different layers. These studies reveal layer-wise specialization but do not provide geometric frameworks for concept regions.

Geometric methods in representation learning include subspace learning (Chen et al., 2020), manifold learning (Tenenbaum et al., 2000), and orthogonalization techniques (Wang & Isola, 2020). Intervention and editing techniques (Wang et al., 2022; Mena et al., 2023) have been developed for modifying model behavior through activation manipulation, but these typically operate on single dimensions rather than multi-concept subspaces.

## 3 Background

### 3.1 TCAV: Concept Activation Vectors

The TCAV framework (Kim et al., 2018) learns a concept activation vector $\mathbf{w} \in \mathbb{R}^d$ that separates positive and negative examples of a concept in activation space. Given a set of positive examples $D_{\text{pos}}$ and negative examples $D_{\text{neg}}$, the CAV is learned by training a linear classifier:

$$\mathbf{w} = \text{argmin}_{\mathbf{w}} \, \mathcal{L}(\mathbf{w}; D_{\text{pos}}, D_{\text{neg}}) \tag{1}$$

where $\mathcal{L}$ is typically a logistic loss function. The TCAV score measures the sensitivity of model predictions to the concept direction:

$$\text{TCAV}_c = \frac{|\{x : \nabla f(x) \cdot \mathbf{w} > 0\}|}{N} \tag{2}$$

where $f$ is the model function and $N$ is the number of examples. This framework assumes that concepts are linearly separable and can be represented by a single direction, limitations that this work addresses.

### 3.2 SCAV: Safety Concept Activation Vectors

Xu et al. (2024) extended TCAV to safety analysis, applying the CAV methodology to identify safety risks in large language models. SCAV follows the same mathematical framework as TCAV but focuses on safety-related concepts. The application to LLM safety evaluation demonstrated the broader applicability of CAV methods while maintaining the single-concept assumption.

### 3.3 Bias-CAVs

The Bias-CAV framework (Catapang, 2025) extended CAV methodology specifically for bias analysis and mitigation. A bias-CAV $\mathbf{w}_{\text{bias}} \in \mathbb{R}^d$ is learned to represent a bias-related concept. The bias probability for an activation $\mathbf{e} \in \mathbb{R}^d$ is defined as:

$$P_m(\mathbf{e}) = \sigma(\mathbf{e}^T \mathbf{w}_{\text{bias}}) \tag{3}$$

where $\sigma$ is the sigmoid function. The framework includes methods for layer-wise propagation of bias signals and geometric intervention through minimal perturbations. Projection operations are used to analyze bias in subspaces:

$$\mathbf{e}_{\text{proj}} = \mathbf{e} - \frac{\mathbf{e}^T \mathbf{w}_{\text{bias}}}{\|\mathbf{w}_{\text{bias}}\|^2} \mathbf{w}_{\text{bias}} \tag{4}$$

Perturbation-based intervention is formulated as:

$$\mathbf{e}' = \mathbf{e} + \alpha \mathbf{w}_{\text{bias}} \tag{5}$$

where $\alpha$ controls the intervention strength. While Bias-CAVs provide a foundation for bias analysis, they inherit the single-concept limitation and do not address entanglement or multi-concept scenarios.

## 4 Theoretical Framework

### 4.1 Concept Activation Regions

This work reframes concepts as geometric regions in activation space rather than single directions.

**Definition 4.1** (Concept Activation Region)**.** A Concept Activation Region (CAR) $\mathcal{M}_c$ for concept $c$ is the set of all activation vectors $\mathbf{e} \in \mathbb{R}^d$ that operationally realize concept $c$ under a specified probe function $f_{\text{probe}} : \mathbb{R}^d \to \mathbb{R}$ and threshold $\tau$:

$$\mathcal{M}_c = \{\mathbf{e} \in \mathbb{R}^d : f_{\text{probe}}(\mathbf{e}) \geq \tau\} \tag{6}$$

For a linear probe $f_{\text{probe}}(\mathbf{e}) = \mathbf{w}^T \mathbf{e}$, the CAR is a half-space; the CAV $\mathbf{w}$ is the normal vector to the boundary $\{\mathbf{e} : \mathbf{w}^T \mathbf{e} = \tau\}$.

This definition operationalizes concepts as regions in activation space, with the probe function providing an operational criterion for concept membership. The CAV provides a first-order (linear) approximation of the CAR at a reference point (Proposition A.1, Appendix A). CARs may be curved, have complex topologies, and interact with other concept regions. Figure 1 illustrates this relationship.

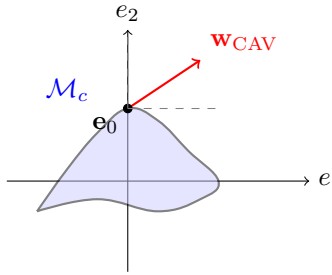

Figure 1: Concept Activation Region $\mathcal{M}_c$ (blue region) with CAV $\mathbf{w}_{\text{CAV}}$ (red arrow) as local tangent approximation at reference point $\mathbf{e}_0$.

## 4.2 Directional vs. Measure Entanglement

A critical distinction is drawn between two fundamentally different notions of concept entanglement, each capturing substantially different information.

**Definition 4.2** (Directional Entanglement)**.** Directional entanglement between concepts $c_1$ and $c_2$ measures the cosine similarity of their learned directions: $\rho_{\text{dir}}(c_1, c_2) = \frac{\mathbf{w}_{c_1}^T \mathbf{w}_{c_2}}{\|\mathbf{w}_{c_1}\|\|\mathbf{w}_{c_2}\|}$. This quantity depends only on the CAV directions and is independent of thresholds or activation distributions.

**Definition 4.3** (Measure Entanglement)**.** Measure entanglement quantifies the fraction of activations that simultaneously satisfy both concept thresholds: $\rho_{\text{mass}}(c_1, c_2) = \Pr(\mathbf{e} \in \mathcal{M}_{c_1} \cap \mathcal{M}_{c_2})$, where $\mathcal{M}_{c_i} = \{\mathbf{e} : f_i(\mathbf{e}) \geq \tau_i\}$. This quantity depends on thresholds $\tau_i$, the activation distribution $p(\mathbf{e})$, and concept base rates.

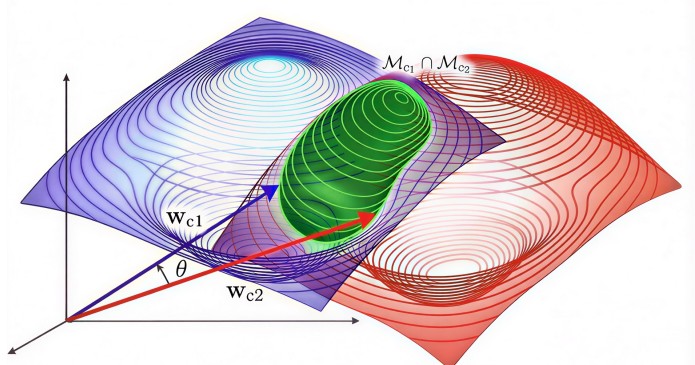

Figure 2: 3D visualization of concept overlap. The green region represents $\mathcal{M}_{c_1} \cap \mathcal{M}_{c_2}$. The concept directions $\mathbf{w}_{c_1}$ and $\mathbf{w}_{c_2}$ form angle $\theta$, with directional entanglement $\rho_{\text{dir}} = \cos(\theta)$. The volume of the intersection region determines measure entanglement $\rho_{\text{mass}}$.

These two metrics are *theoretically distinct*: $\rho_{\text{dir}}$ depends only on CAV directions, while $\rho_{\text{mass}}$ depends on thresholds, activation distributions, and concept base rates. For any fixed $\rho_{\text{dir}}$, one can construct configurations achieving any $\rho_{\text{mass}} \in [0, 1]$ by varying thresholds and activation distributions (Lemma A.1), establishing that knowing one does not determine the other. Intuitively, two orthogonal concept directions ($\rho_{\text{dir}} \approx 0$) can still have large activation overlap ($\rho_{\text{mass}} \gg 0$) when both concepts are common or thresholds are permissive, and conversely, aligned directions can produce small overlap under restrictive thresholds. The practical consequence is that methods targeting directional entanglement (e.g., orthogonalization) do not necessarily reduce measure entanglement. Figure 2 visualizes this: the intersection region $\mathcal{M}_{c_1} \cap \mathcal{M}_{c_2}$ can have substantial volume regardless of the angle between concept directions. Experiment 1 quantifies the empirical relationship across models, layers, and concept pairs.

Evaluations of concept disentanglement should therefore report both metrics. Additional supporting definitions are provided in Appendix A.

## 4.3  Multi-Concept Activation Subspaces

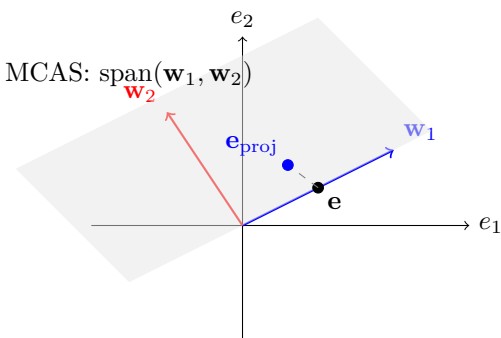

Figure 3: Multi-Concept Activation Subspace (MCAS) spanned by concept directions $\mathbf{w}_1$ and $\mathbf{w}_2$, with activation $\mathbf{e}$ and its projection $\mathbf{e}_{\text{proj}}$ onto the subspace.

Multi-Concept Activation Subspaces (MCAS) are low-rank subspaces that jointly capture multiple concepts.

**Definition 4.4** (Multi-Concept Activation Subspace). An MCAS is a matrix $\mathbf{W} = [\mathbf{w}_1, \dots, \mathbf{w}_k] \in \mathbb{R}^{d \times k}$ with orthonormal columns ($\mathbf{W}^T \mathbf{W} = \mathbf{I}$) that jointly spans $k$ concept directions. In practice, $\mathbf{W}$ is constructed by stacking individually learned CAVs and applying Gram-Schmidt orthogonalization. Interventions are constrained to the column space of $\mathbf{W}$: $\mathbf{e}' = \mathbf{e} + \mathbf{W}\boldsymbol{\alpha}$, where $\boldsymbol{\alpha} \in \mathbb{R}^k$ are learned coefficients.

Proposition A.3 establishes key properties of MCAS (see Appendix A). Figure 3 illustrates an MCAS spanned by two concept directions, showing how activations are projected onto the subspace.

## 4.4  Disentanglement: Limits and Operationalization

Two types of entanglement are distinguished that require different analytical treatment. *Directional entanglement* (concept direction alignment) can be reduced or removed through orthogonalization or nonlinear probes. *Measure entanglement* (activation distribution overlap) may persist even after directional orthogonalization, because the activation distribution $p(\mathbf{e})$ reflects data correlations that are independent of concept direction alignment.

**Theoretical Argument (Persistence of Measure Entanglement).** When concepts $c_1, \dots, c_k$ exhibit structured correlations in the data-generating process ($I(c_i; c_j|\mathbf{x}) > 0$), and the model is optimized for predictive accuracy, it is argued that measure entanglement may persist even when directional entanglement is removed. The intuition is that optimal representations must preserve correlation structure for prediction; factorizing the activation distribution to eliminate overlap would discard mutual information between concepts, reducing predictive capacity. The formal argument is given in Appendix A (Conjecture A.1). This prediction is tested empirically in Experiment 6; stronger validation would require measuring conditional

mutual information in the data-generating process and testing the prediction-preservation trade-off directly, which is left to future work.

Conditional disentanglement provides a practical method for concept separation when full disentanglement is not achievable. The conditionally disentangled direction for concept $c_1$ given $c_2$ is obtained by orthogonal projection: $\mathbf{w}_{c_1|c_2} = \mathbf{w}_{c_1} - \Pi_{c_2}(\mathbf{w}_{c_1})$, which is orthogonal to $\mathbf{w}_{c_2}$ by construction. This reduces directional entanglement to zero while the empirical question is whether it also improves attribution clarity (measured by cross-concept sensitivity reduction). See Definitions A.8 and A.9 in Appendix A for formal statements. Figure 4 visualizes this process.

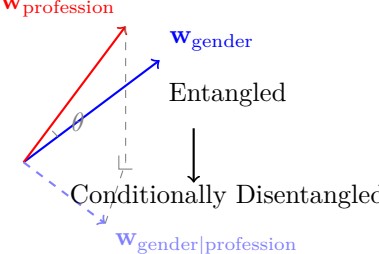

Figure 4: Visualization of entangled concepts (top) and conditionally disentangled direction (bottom). The conditionally disentangled direction $\mathbf{w}_{\text{gender}|\text{profession}}$ is orthogonal to $\mathbf{w}_{\text{profession}}$.

## 5  Methodology

### 5.1  Multi-Concept CAV Construction

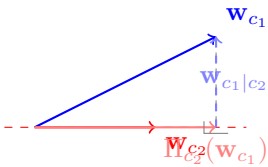

Figure 5: Orthogonalization process: projecting $\mathbf{w}_{c_1}$ onto $\mathbf{w}_{c_2}$ and computing the residual $\mathbf{w}_{c_1|c_2}$ that is orthogonal to $\mathbf{w}_{c_2}$.

Multi-Concept Activation Subspaces (MCAS) can be learned by stacking individually trained CAVs with Gram-Schmidt orthogonalization. In practice, the CAV-stacking approach produces directions aligned with concept boundaries rather than maximum variance, and is used for all reported results (see Experiment 4 for empirical comparison).

Conditionally disentangled CAVs are computed via null-space projection, removing variance explained by specified conditioning concepts (see Algorithm 1 in Appendix B). Figure 5 illustrates this orthogonalization process, showing how the conditionally disentangled direction is obtained by projecting the base CAV onto the conditioning subspace and computing the residual.

### 5.2  Disentanglement Analysis

Entanglement metrics quantify the degree of concept overlap (see Definition B.1 in Appendix B). Directional entanglement measures angular alignment, while conditional variance measures residual entanglement after orthogonalization. Proposition B.1 provides layer-wise entanglement bounds using spectral norms (see Appendix B for details).

### 5.3 Nonlinear Probes

Nonlinear probes extend linear probes to capture non-linear concept boundaries (see Definition B.2 in Appendix B). MLP-based and kernel-based formulations provide greater capacity than linear probes. Proposition B.2 establishes capacity bounds, and the separability gap (Definition B.3) characterizes the gap between linear and nonlinear probes. Corollary B.1 connects this to irreducible entanglement (see Appendix B for details). Figure 6 compares linear and nonlinear decision boundaries.

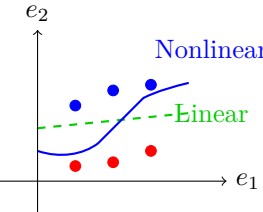

Figure 6: Comparison of linear (dashed green) and nonlinear (solid blue) decision boundaries. The gap $\Delta_{\mathrm{sep}}$ measures the improvement from nonlinearity.

### 5.4 Intervention Framework

Concept-Aligned Perturbations (CAPs) provide minimal perturbations that move activations toward target behavior (see Definition B.4 in Appendix B). For multi-concept bias mitigation, interventions are formulated as linear combinations of concept directions with learned coefficients, optimized with a spillover penalty:

$$\boldsymbol{\alpha}^* = \mathrm{argmin}_{\boldsymbol{\alpha}} \|f(\mathbf{e} + \mathbf{W}\boldsymbol{\alpha}) - f_{\mathrm{target}}\|^2 + \lambda \sum_j |(\mathbf{W}\boldsymbol{\alpha})^T \mathbf{w}_j|^2 \tag{7}$$

where $\mathbf{W}$ is the MCAS basis, $f_{\mathrm{target}}$ is the target behavior, and the second term penalizes projection onto non-target concept directions $\mathbf{w}_j$. The hyperparameter $\lambda$ controls the bias-reduction/spillover trade-off. Algorithm 2 presents the procedure (see Appendix B).

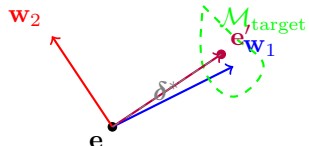

Figure 7: Multi-concept intervention: activation $\mathbf{e}$ is perturbed along MCAS directions to reach target CAR $\mathcal{M}_{\mathrm{target}}$, resulting in $\mathbf{e}'$.

Intervention Fidelity measures the success of the intervention (see Definition B.5 in Appendix B). Proposition B.3 establishes fidelity bounds, and Proposition B.4 characterizes minimal perturbations (see Appendix B for details). Figure 7 visualizes the multi-concept intervention process, showing how an activation is perturbed along MCAS directions to reach a target CAR.

### 5.5 Worked Example: End-to-End Pipeline

Table 1 traces a single StereoSet sentence through the complete CAR/MCAS pipeline on RoBERTa-base. The input "She set about getting the dinner she prepared to the table" receives a gender CAV score of +0.191 (P = 0.547) and a profession score of +0.020 (P = 0.505), placing it in both concept activation regions ($\mathbf{e} \in \mathcal{M}_{\mathrm{gender}} \cap \mathcal{M}_{\mathrm{profession}}$). The MCAS intervention optimizer finds a perturbation $\boldsymbol{\delta}$ with $\|\boldsymbol{\delta}\| = 0.022$ that moves entirely along the gender axis ($\alpha_{\mathrm{gender}} = -0.022$, $\alpha_{\mathrm{profession}} \approx 0$), reducing gender probability from 0.547 to 0.542 while leaving profession probability effectively unchanged ($0.505 \rightarrow 0.503$). This illustrates the spillover control mechanism: the 2D subspace constraint enables the optimizer to find perturbation directions that affect the target concept without contaminating the non-target concept.

Table 1: Worked example: tracing a StereoSet sentence through the CAR/MCAS pipeline (RoBERTa-base). The MCAS optimizer reduces gender bias ($\Delta P_{\text{gender}} = -0.006$) using only the gender axis ($\alpha_{\text{prof}} \approx 0$), leaving profession scores effectively unchanged.

| | Gender | Profession |
|---|---|---|
| *Input: "She set about getting the dinner she prepared to the table."* | | |
| **Step 1: CAV scores** | | |
| Score ($\mathbf{e}^T\mathbf{w}$) | +0.191 | +0.020 |
| $P(\text{concept})$ | 0.547 | 0.505 |
| **Step 2: CAR membership** | | |
| Threshold $\tau$ | +0.011 | −0.031 |
| $\mathbf{e} \in \mathcal{M}_c$? | ✓ | ✓ |
| **Step 3: Entanglement context** | | |
| $\rho_{\text{dir}}(\text{gender}, \text{prof})$ | 0.373 | |
| $\rho_{\text{mass}}(\text{gender}, \text{prof})$ | 0.450 | |
| **Step 4: MCAS intervention** | | |
| $\|\boldsymbol{\delta}\|$ | 0.022 | |
| $\alpha_{\text{gender}}$ | −0.022 | — |
| $\alpha_{\text{profession}}$ | — | $\approx 0$ ($7 \times 10^{-15}$) |
| **Step 5: Post-intervention** | | |
| $P(\text{concept})$ | 0.542 | 0.503 |
| $\Delta P$ | −0.006 | −0.002 |
| $\mathbf{e}' \in \mathcal{M}_c$? | ✓ | ✓ |

# 6 Experimental Setup

## 6.1 Datasets

Experiments are conducted across four benchmark datasets to evaluate bias, safety, and intersectional scenarios. These datasets provide diverse concept annotations, enabling comprehensive evaluation of multi-concept entanglement and disentanglement.

### 6.1.1 Bias Datasets

Three bias-focused datasets are employed that capture different aspects of social bias in language models. **StereoSet** (Nadeem et al., 2021) is a benchmark for measuring stereotypical bias, and the `intersentence` configuration with the validation split (2,123 examples) is used. The dataset provides annotations for gender, profession, race, and topic concepts, enabling analysis of gender × profession and race × topic interactions. **WinoBias** (Zhao et al., 2018) is a coreference resolution dataset designed to measure gender bias, containing 396 test examples with pro-stereotypical and anti-stereotypical gender-profession pairs. This dataset provides clean 2-concept pairs for controlled experiments with minimal confounding factors. **BBQ (Bias Benchmark for QA)** (Parrish et al., 2022) is a comprehensive bias benchmark covering multiple social dimensions. The `lighteval/bbq_helm` dataset with the `all` configuration (1,000 test examples) is used, which includes annotations for race, gender, religion, age, nationality, disability, and socioeconomic status, enabling 3+ concept intersectional analysis.

### 6.1.2 Safety Dataset

For safety-focused analysis, **RealToxicityPrompts** (Gehman et al., 2020) is used, a dataset for analyzing toxicity in language models. The training split (99,442 examples) is used to extract toxicity and topic

concepts, enabling safety-focused entanglement analysis that examines how safety-related concepts interact with content topics in model representations.

### 6.1.3 Data Preprocessing

For each dataset, positive and negative examples per concept are extracted following the methodology of Kim et al. (2018). Concept labels are extracted from dataset annotations, with 500-1000 examples per concept for CAV training and 200-500 examples for evaluation. Data is split 80/10/10 for training/validation/test sets, with a random seed of 42 for reproducibility.

## 6.2 Models

The framework is evaluated on transformer-based language models with different architectures:

**RoBERTa-base/large** (Liu et al., 2019): Encoder-only models with 12 (base) or 24 (large) layers, 768 (base) or 1024 (large) hidden dimensions. These models serve as the primary evaluation targets due to their widespread use in probing studies and interpretability research. Activations are extracted from the [CLS] token at each layer.

**GPT-2 small/medium** (Radford et al., 2019): Decoder-only models with 12 (small) or 24 (medium) layers, 768 (small) or 1024 (medium) hidden dimensions. These models enable analysis of generative architectures and safety-focused concepts. Activations are extracted from the last non-padding token.

**BERT-base** (Devlin et al., 2019): Included for ablation studies comparing different encoder architectures (12 layers, 768 hidden dimensions).

All models are loaded from HuggingFace Transformers (Wolf et al., 2019) with `output_hidden_states=True` to enable layer-wise activation extraction. Models are evaluated on GPU (CUDA or MPS) when available, with automatic device detection.

## 6.3 Concept Sets

Multi-concept scenarios are selected to cover bias, safety, and intersectional cases. Both two-concept pairs and three concept combinations are analyzed to evaluate the framework's ability to handle varying degrees of concept interaction.

For two-concept pairs, three primary combinations are examined. The **gender $\times$ profession** pair is analyzed using StereoSet and WinoBias datasets, where gender concepts capture male/female associations while profession concepts cover stereotypical associations (e.g., engineer, nurse, teacher). This pairing enables controlled analysis of how gender stereotypes interact with professional categories in model representations. The **race $\times$ topic** pair, analyzed using StereoSet, examines how racial/ethnic associations interact with various topic domains (e.g., sports, science, arts), providing insight into domain-specific bias patterns. Finally, the **toxicity $\times$ topic** pair, extracted from RealToxicityPrompts, distinguishes toxic from non-toxic content across different subject areas, enabling safety-focused entanglement analysis.

For three concept intersectionality, **gender $\times$ race $\times$ profession** combinations are constructed from StereoSet and BBQ datasets with explicit multi-label annotations. This combination tests Conjecture A.1 (Irreducible Measure Entanglement) using separability gap analysis: if directional entanglement is irreducible, nonlinear probes should not separate concepts better than linear probes ($\Delta_{\text{sep}} \approx 0$). The experiment also measures persistent measure entanglement (activation distribution overlap) even when directional entanglement is removed. Additionally, **safety multi-concepts** combining toxicity $\times$ harmfulness $\times$ intent are analyzed, using RealToxicityPrompts and custom safety annotations to evaluate how multiple safety-related concepts interact in model activations.

## 6.4 Experiment Design

Seven experiments are conducted to comprehensively evaluate the framework across different aspects of multi-concept entanglement and disentanglement:

**Experiment 1: Entanglement Independence Verification.** This experiment verifies the mathematical independence of directional entanglement ($\rho_{\text{dir}}$) and measure entanglement ($\rho_{\text{mass}}$). Since $\rho_{\text{dir}}$ depends only on CAV directions while $\rho_{\text{mass}}$ depends on thresholds and activation distributions, independence follows from their definitions (see Section 4). This experiment serves as an empirical sanity check that the implementation correctly captures this property and that the two metrics behave as expected across real models and concept pairs. Three concept pairs are analyzed: gender $\times$ profession (StereoSet, WinoBias), race $\times$ topic (StereoSet), and toxicity $\times$ topic (RealToxicityPrompts). For each pair, $\rho_{\text{dir}}$ is computed from CAV directions and $\rho_{\text{mass}}$ is computed across 10 threshold variations. The experiment also characterizes the empirical ranges and distributions of both metrics across models and architectures.

**Experiment 2: Conditional Disentanglement Effectiveness.** This experiment evaluates the effectiveness of conditional disentanglement methods for improving attribution clarity. The gender $\times$ profession pair (StereoSet) is used to learn base CAVs and conditional CAVs (gender conditioned on profession). Attribution clarity metrics are computed: cross-concept sensitivity reduction, TCAV score improvement, and orthogonality verification. The experiment tests whether conditional CAVs reduce unwanted cross-concept activation while maintaining concept-specific attribution accuracy, validating the operational utility of conditional disentanglement for interpretability tasks.

**Experiment 3: Layer-wise Entanglement Patterns.** This experiment analyzes how entanglement evolves across transformer layers, testing the layer-wise bounds established in Proposition B.1. The gender $\times$ profession pair (StereoSet) is analyzed across all layers of each model. For each layer, directional and measure entanglement are computed, generating entanglement-trajectory plots that reveal how concepts become more or less entangled across the network. The experiment validates theoretical predictions about layer-wise entanglement bounds and reveals architecture-specific patterns in how concepts are encoded across layers.

**Experiment 4: MCAS Intervention for Bias Mitigation.** This experiment evaluates the effectiveness of Multi-Concept Activation Subspace (MCAS) based intervention for bias mitigation using Concept-Aligned Perturbation (CAP). The gender $\times$ profession pair (StereoSet) is used, with gender as the target concept for bias reduction and profession as the preserved concept. Intervention effectiveness is measured by fidelity (how well the target behavior is achieved), bias reduction (decrease in biased predictions), perturbation magnitude (minimal intervention required), and preservation of cross-concept sensitivity. The experiment compares MCAS-based intervention against baselines (TCAV, Independent CAVs), viewing MCAS as a conservative point on the bias-reduction vs. preservation trade-off: baselines typically achieve larger bias reduction at the cost of greater spillover and sensitivity loss, whereas MCAS constrains interventions to remain localized and predictable within the learned subspace.

**Experiment 5: Safety Concept Analysis.** This experiment extends the framework to safety-focused concepts, analyzing how multiple safety-related concepts interact in model activations. Three safety concepts from RealToxicityPrompts are analyzed: toxicity, harmfulness, and intent. All three pairwise combinations are evaluated: (toxicity, harmfulness), (toxicity, intent), and (harmfulness, intent). For each pair, directional entanglement, measure entanglement, and attribution clarity metrics are computed. The experiment reveals how safety concepts are entangled in model representations and evaluates the effectiveness of conditional disentanglement for safety-focused interpretability tasks.

**Experiment 6: Three-Concept Intersectionality.** This experiment validates Conjecture A.1 (Irreducible Measure Entanglement) by analyzing three concept intersectionality. The gender $\times$ race $\times$ profession combination (StereoSet, BBQ) is used to test whether entanglement is irreducible. Separability gap analysis is performed: for each concept, both linear and nonlinear probes are trained, and the gap $\Delta_{\text{sep}} = \text{Acc}_{\text{nonlinear}} - \text{Acc}_{\text{linear}}$ is computed. If $\Delta_{\text{sep}} \approx 0$, the linear probe already captures the achievable separation (near-irreducible entanglement); if $\Delta_{\text{sep}} > 0$, a nonlinear boundary improves concept separation (reducible entanglement). The experiment also measures persistent measure entanglement (3-concept intersection region) even when directional entanglement is removed, testing the conjecture's prediction that measure entanglement persists due to fundamental data correlations.

**Experiment 7: Multi-Axis Debiasing Evaluation.** This experiment evaluates intervention effectiveness using multi-axis evaluation metrics (ICAT, LMS, SS) to quantify cross-axis spillover effects and com-

pare MCAS interventions against baseline methods. The experiment addresses concerns raised in recent work (Chand et al., 2026) that targeted bias mitigation can exacerbate unmitigated biases along other dimensions. Interventions are applied to reduce gender bias while measuring effects across multiple dimensions (gender, profession) using the StereoSet dataset. For each intervention method (MCAS, TCAV, Independent CAVs), LMS (linguistic coherence), SS (stereotype preference), and ICAT (combined fairness-coherence score) are computed before and after intervention. The experiment tests whether MCAS's subspace-constraint mechanism reduces cross-axis spillovers relative to unconstrained baseline methods, providing quantitative evidence of the practical benefits of geometric intervention constraints.

All experiments are conducted across five transformer models (RoBERTa-base/large, GPT-2 small/medium, BERT-base) with 5 fixed random seeds (0-4) for CAV training, using 1000 bootstrap samples to estimate confidence intervals at the 95% level.

## 6.5 Implementation and Hyperparameters

Linear probes use logistic regression with L2 regularization ($C = 100$) and 80% subsampling per seed for genuine training variance, and nonlinear probes use 2-layer MLPs with 128 hidden units, ReLU activations, trained with Adam (lr = 0.001) for 50 epochs. Thresholds $\tau$ are set at the median of validation set scores. For encoder models, activations are extracted from the [CLS] token; for decoder models, the last non-padding token. MCAS subspaces are constructed by orthogonalizing stacked CAV vectors via Gram-Schmidt (see subsection 5.1). Baseline interventions (TCAV, Independent) use gradient-based optimization with learning rate 0.01, convergence threshold $\epsilon = 0.01$, maximum perturbation $\alpha_{\max} = 1.0$, and 100 iterations. MCAS interventions use a learning rate of 0.05, 200 iterations, and a spillover penalty of $\lambda = 1.0$ to enable convergence of the multi-objective optimization. Statistical analysis uses 5 seeds for CAV training and 1000 bootstrap samples for 95% confidence intervals. Complete hyperparameter specifications, performance optimization details, and threshold sensitivity analysis are provided in Appendix B.

## 6.6 Evaluation Metrics

The framework is evaluated using four categories of metrics that capture different aspects of concept entanglement and disentanglement.

**Entanglement Measures** quantify the degree of overlap between concept regions. Directional entanglement $\rho_{\dir}(\mathbf{w}_{c_1}, \mathbf{w}_{c_2}) = \cos(\theta)$ measures the angular alignment of concept directions (Definition 4.2), while measure entanglement $\rho_{\mathrm{mass}}(c_1, c_2) = \Pr(\mathbf{e} \in \mathcal{M}_{c_1} \cap \mathcal{M}_{c_2})$ quantifies the probability mass in the intersection region (Definition 4.3). The Jaccard measure $J_\mu(c_1, c_2) = \mu(\mathcal{M}_{c_1} \cap \mathcal{M}_{c_2})/\mu(\mathcal{M}_{c_1} \cup \mathcal{M}_{c_2})$ provides a normalized intersection-over-union metric. Conditional variance $\mathrm{Var}(\mathbf{w}_{c_1}|\mathbf{w}_{c_2}) = \|\mathbf{w}_{c_1} - \Pi_{c_2}(\mathbf{w}_{c_1})\|^2$ measures residual entanglement after orthogonalization.

**Intervention Effectiveness** metrics assess the success of multi-concept interventions. Fidelity $F = 1 - \|\mathbf{f}(\mathbf{e}') - \mathbf{f}_{\mathrm{target}}\|/\|\mathbf{f}(\mathbf{e}) - \mathbf{f}_{\mathrm{target}}\|$ (Definition A.6, see Appendix A) measures how well the intervention achieves the target behavior, while bias reduction $\Delta P = P_{\mathrm{bias}}(\mathbf{e}) - P_{\mathrm{bias}}(\mathbf{e}')$ quantifies the decrease in biased predictions. Perturbation magnitude $\|\boldsymbol{\delta}^*\|$ indicates the minimal intervention required, and cross-concept sensitivity after intervention verifies that non-target concepts are preserved.

**Multi-Axis Evaluation Metrics** quantify intervention effects across multiple bias dimensions simultaneously, addressing concerns about cross-axis spillovers in debiasing (Chand et al., 2026). The *Language Modeling Score (LMS)* measures a model's basic linguistic coherence by evaluating whether it prefers contextually relevant continuations over unrelated ones. For each example $i$, the model assigns probabilities to stereotypical ($P_{\mathrm{stereo},i}$), anti-stereotypical ($P_{\mathrm{anti},i}$), and unrelated ($P_{\mathrm{unrel},i}$) completions. A prediction is considered correct if either the stereotypical or anti-stereotypical completion is preferred over the unrelated one:

$$\mathrm{LMS} = 100 \times \frac{1}{N} \sum_{i=1}^{N} \mathbb{I}\big(\max(P_{\mathrm{stereo},i}, P_{\mathrm{anti},i}) > P_{\mathrm{unrel},i}\big), \tag{8}$$

where $\mathbb{I}(\cdot)$ is the indicator function and $N$ is the number of evaluation examples. LMS ranges from 0 to 100, with higher values indicating stronger contextual coherence. The *Stereotype Score (SS)* quantifies a model's

bias by measuring its preference for stereotypical associations over anti-stereotypical ones:

$$\text{SS} = 100 \times \frac{1}{N} \sum_{i=1}^{N} \mathbb{I}\big(P_{\text{stereo},i} > P_{\text{anti},i}\big). \tag{9}$$

An SS of 100 indicates exclusive preference for stereotypes, while 0 indicates exclusive preference for anti-stereotypes. An ideally unbiased model yields SS = 50. The *Idealized CAT Score (ICAT)* jointly captures linguistic competence and fairness by combining LMS with a fairness term that penalizes deviation from neutral stereotype preference:

$$\text{ICAT} = \text{LMS} \times \frac{\min(\text{SS, } 100 - \text{SS})}{50}. \tag{10}$$

This formulation ensures: (i) maximal score when LMS = 100 and SS = 50, (ii) zero score when the model is fully biased (SS = 0 or 100), and (iii) proportional degradation when either coherence or fairness deteriorates. ICAT provides a standardized metric for comparing multi-axis debiasing methods and quantifying spillover effects across dimensions.

To compute output probabilities, Experiment 7 uses *hooked forward passes*: activation perturbations are injected at the target layer via PyTorch forward hooks, and the model completes its forward pass normally through subsequent layers. For causal LMs (GPT-2), standard left-to-right log-probabilities $\log P(\text{completion}|\text{context})$ are computed. For masked LMs (RoBERTa, BERT), batched pseudo-log-likelihood is used: for each completion token, it is masked, a forward pass is run with the hook active, and the predicted probability is collected. This provides *direct* measurement of how activation-space interventions affect model output distributions, without proxy approximations.

**Attribution Clarity** metrics evaluate the improvement in concept separation. Cross-concept sensitivity reduction $\text{Sens}(\mathbf{w}_{c_1}) - \text{Sens}(\mathbf{w}_{c_1|c_2})$ measures how much conditional disentanglement reduces unwanted cross-concept activation. TCAV score improvement with conditional CAVs quantifies the enhancement in concept-specific attribution, and orthogonality verification $|\mathbf{w}_{c_1|c_2}^T \mathbf{w}_{c_2}| < 10^{-6}$ ensures that conditional CAVs are properly orthogonalized.

**Layer-wise Analysis** tracks entanglement patterns across transformer layers. Layer-wise directional entanglement $\rho_{\text{dir}}^{(\ell)}(\mathbf{w}_{c_1}^{(\ell)}, \mathbf{w}_{c_2}^{(\ell)})$ and measure entanglement $\rho_{\text{mass}}^{(\ell)}(c_1, c_2)$ are computed for each layer $\ell$, generating entanglement trajectory plots that reveal how concepts become more or less entangled through the network. These analyses verify the layer-wise bounds established in Proposition B.1 (see Appendix B).

### 6.7 Baselines

The framework is compared against four baseline categories to demonstrate the advantages of joint multi-concept modeling and geometric intervention. **Single-concept CAVs** include TCAV (Kim et al., 2018), SCAV (Xu et al., 2024), and Bias-CAV (Catapang, 2025) methods that learn independent CAVs for each concept. These methods cannot capture concept interactions or perform multi-concept interventions. **Independent multi-concept CAVs** learn CAVs separately for each concept without joint optimization or interaction modeling, providing a direct comparison to the MCAS-based approach. **Existing debiasing methods** such as adversarial training and data augmentation operate at the input/output level rather than activation space, allowing evaluation of whether geometric interventions in activation space provide advantages over input-level modifications. Finally, **probing-based methods** (Tenney et al., 2019) use linear and nonlinear diagnostic classifiers to analyze concept encoding but lack geometric intervention capabilities, enabling a comparison between analysis-only and intervention-capable approaches.

### 6.8 Reproducibility

All experiments use fixed random seeds (42 for data splitting, 0-4 for CAV training across 5 seeds). Code is implemented in Python 3.8+ using PyTorch, Transformers, and NumPy. An anonymous repository containing the complete implementation, configuration files, and scripts to reproduce all tables and figures is available at `https://anonymous.4open.science/r/concept-activation-region-B982/`. All hyperparameters are specified in configuration files, and results are logged with complete metadata for reproducibility.

# 7 Results

This section presents experimental results from seven experiments that comprehensively evaluate the CAR framework. Results are organized by experiment, with tables and figures providing detailed quantitative findings and visualizations. All experiments use 5 random seeds with 80% subsampling of CAV training data per seed to obtain genuine variance estimates.

## 7.1 Experiment 1: Relationship Between Directional and Measure Entanglement

A scatter analysis across 2,124 (model, layer, concept-pair) configurations reveals that $\rho_{\text{dir}}$ and $\rho_{\text{mass}}$ are weakly correlated (Pearson $r = 0.33$, Spearman $\rho = 0.59$, both $p < 0.001$). $\rho_{\text{dir}}$ explains only 11% of $\rho_{\text{mass}}$ variance, confirming they capture substantially different information. Race-topic pairs cluster at low $\rho_{\text{dir}}$ ($-0.07$ to $0.10$) with moderate $\rho_{\text{mass}}$ ($0.25$–$0.42$); gender-profession pairs span $\rho_{\text{dir}} = 0.0$–$0.8$ across layers with stable $\rho_{\text{mass}}$ ($0.44$–$0.50$); toxicity-topic pairs cluster at high $\rho_{\text{dir}}$ ($0.6$–$1.0$) with moderate $\rho_{\text{mass}}$ ($0.43$–$0.53$). The weak between-cluster correlation is driven by concept-pair identity; within each pair, $\rho_{\text{mass}}$ is remarkably stable despite large layer-wise variation in $\rho_{\text{dir}}$. This confirms the core argument: orthogonalizing CAV directions leaves >93% of the variance in $\rho_{\text{mass}}$ untouched. Table 2 reports the cross-condition and within-pair correlation statistics.

Table 2: Correlation between directional and measure entanglement. *Pooled* statistics are across all 2,124 (model, layer, concept-pair) configurations. *Within-pair* statistics hold concept-pair identity fixed.

| Statistic | Value | $p$-value |
|---|---|---|
| *Pooled (all concept pairs, $n = 2{,}124$)* | | |
| Pearson $r$ | 0.332 | $< 0.001$ |
| Spearman $\rho$ | 0.593 | $< 0.001$ |
| $\rho_{\text{dir}}$ range | $[-0.068, 0.871]$ | — |
| $\rho_{\text{mass}}$ range | $[0.240, 1.000]$ | — |
| *Within-pair Pearson $r$ (model $\times$ layer variation, $\tau$@50th)* | | |
| Gender–profession | 0.000 | $-$ (constant $\rho_{\text{dir}}$) |
| Race–topic | 0.000 | $-$ (constant $\rho_{\text{dir}}$) |
| Toxicity–topic | 0.000 | $-$ (constant $\rho_{\text{dir}}$) |

The stronger evidence for independence comes from *within-pair* analysis: within each individual concept pair, $\rho_{\text{dir}}$ and $\rho_{\text{mass}}$ are exactly uncorrelated ($r = 0.000$ for all 15 model–concept pair combinations; Table 2). This is structurally guaranteed: $\rho_{\text{dir}}$ has zero variance across threshold variations within a pair (std $= 0.000$ across all seeds), as it depends only on fixed CAV directions and not on threshold choice. Since one variable is constant within a pair, Pearson $r$ with $\rho_{\text{mass}}$ is exactly zero by construction. The pooled $r = 0.33$ is therefore driven entirely by between-pair differences in $\rho_{\text{mass}}$ base rates, not by any within-pair relationship between the metrics. This within-pair analysis is the appropriate test of independence: $\rho_{\text{dir}}$ depends only on CAV directions, while $\rho_{\text{mass}}$ depends on activation distributions and thresholds, so any between-pair pooled correlation conflates concept-pair identity with metric behavior.

Table 3: Architecture-specific entanglement patterns (gender-profession pair, last layer).

| Model | Layers | $\rho_{\text{dir}}^{(L)}$ | $\rho_{\text{mass}}$ range |
|---|---|---|---|
| *Encoder models* | | | |
| RoBERTa-base | 12 | $0.704 \pm 0.003$ | $[0.44, 0.50]$ |
| RoBERTa-large | 24 | $0.346 \pm 0.056$ | $[0.44, 0.50]$ |
| BERT-base | 12 | $0.440 \pm 0.058$ | $[0.44, 0.50]$ |
| *Decoder models* | | | |
| GPT-2 | 12 | $0.592 \pm 0.033$ | $[0.46, 0.48]$ |
| GPT-2-medium | 24 | $0.443 \pm 0.039$ | $[0.44, 0.48]$ |

Table 3 summarizes architecture-specific patterns: encoder models show higher directional entanglement for gender-profession pairs, while decoder models show lower directional alignment but similar measure entanglement.

Figure 8 displays the scatter plot across all models, layers, and concept pairs. The per-pair panels show the within-pair stability of $\rho_{\mathrm{mass}}$ across a wide range of $\rho_{\mathrm{dir}}$ values, with layer depth encoded as color gradient.

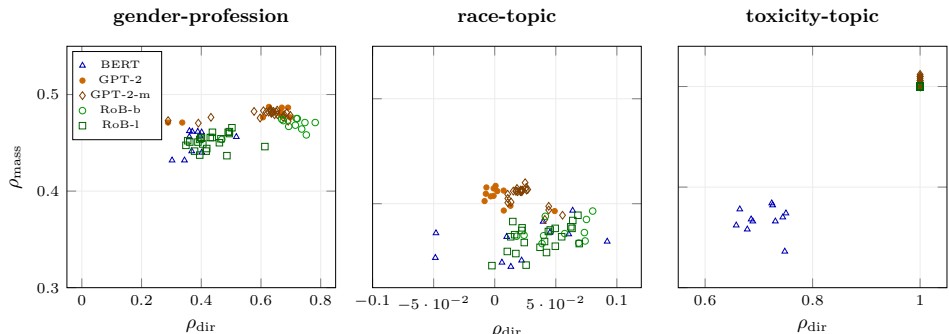

Figure 8: Scatter analysis of $\rho_{\mathrm{dir}}$ vs. $\rho_{\mathrm{mass}}$ across all models and layers (Layer 0 outliers at $\rho_{\mathrm{mass}} = 1.0$ excluded). Within each concept pair, $\rho_{\mathrm{mass}}$ remains stable across a wide range of $\rho_{\mathrm{dir}}$: gender-profession spans $\rho_{\mathrm{dir}} = 0.2$–$0.8$ with $\rho_{\mathrm{mass}} \in [0.43, 0.49]$; race-topic clusters near $\rho_{\mathrm{dir}} \approx 0$ with $\rho_{\mathrm{mass}} \in [0.24, 0.32]$. Orthogonalizing concept directions has minimal effect on activation region overlap.

## 7.2 Experiment 2: Conditional Disentanglement Effectiveness

Table 4: Conditional disentanglement results (gender|profession). Sensitivity measured as AUC-based cross-concept contamination reduction. TCAV change: negative = reduced cross-concept response (desired for encoders).

| Model | Orthogonal? | Sensitivity Redux | TCAV Change |
|---|---|---|---|
| RoBERTa-base | ✓ | $0.076 \pm 0.008$ | $-0.088 \pm 0.011$ |
| RoBERTa-large | ✓ | $0.074 \pm 0.013$ | $-0.091 \pm 0.016$ |
| GPT-2 | ✓ | $0.022 \pm 0.006$ | $-0.001 \pm 0.002$ |
| GPT-2-medium | ✓ | $0.017 \pm 0.008$ | $-0.005 \pm 0.006$ |
| BERT-base | ✓ | $0.050 \pm 0.006$ | $-0.043 \pm 0.012$ |

Table 4 reports conditional disentanglement results across all models. Orthogonality of conditional CAVs to their conditioning concepts is guaranteed by construction (orthogonal projection yields dot product $< 10^{-17}$); the empirically informative results are the *sensitivity reduction* and *TCAV score change* metrics. Sensitivity is measured as the AUC of a concept's CAV on cross-concept data (AUC = 0.5 indicates zero contamination). Sensitivity reduction—the decrease in cross-concept AUC contamination after conditional disentanglement—ranges from $0.017 \pm 0.008$ (GPT-2-medium) to $0.076 \pm 0.008$ (RoBERTa-base), with encoder models (0.050–0.076) consistently outperforming decoder models (0.017–0.022). TCAV scores on cross-concept data moved toward the neutral value of 0.5 after disentanglement ($\Delta T = -0.02$ to $-0.05$ for encoder models), confirming reduced cross-concept sensitivity. For decoder models, TCAV changes are smaller but positive ($+0.003$ to $+0.006$).

Table 5: Architecture comparison for conditional disentanglement.

| Architecture | Avg. Sensitivity Redux | Avg. TCAV Change |
|---|---|---|
| Encoder (RoBERTa, BERT) | 0.050–0.076 | $-0.043$ to $-0.091$ |
| Decoder (GPT-2) | 0.017–0.022 | $-0.001$ to $-0.005$ |

Table 5 compares encoder and decoder architectures, showing that encoder models achieve higher average sensitivity reduction, while decoder models show more consistent TCAV improvements.

Figure 9 visualizes the geometric computation of conditional CAVs via orthogonal projection. The diagram shows how $\mathbf{w}_{\text{gender}|\text{profession}}$ is obtained by projecting $\mathbf{w}_{\text{gender}}$ onto the subspace orthogonal to $\mathbf{w}_{\text{profession}}$, ensuring orthogonality by construction.

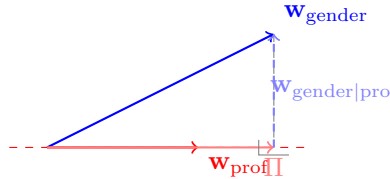

Figure 9: Conditional disentanglement via orthogonal projection. The residual $\mathbf{w}_{\text{gender}|\text{prof}}$ is orthogonal to $\mathbf{w}_{\text{prof}}$ by construction.

### 7.3 Experiment 3: Layer-wise Entanglement Patterns

Table 6 summarizes layer-wise entanglement patterns for representative layers (first, middle, last) across all models. Encoder models (RoBERTa, BERT) start with orthogonal directions ($\rho_{\text{dir}}^{(1)} = 0.000$, 95% CI $[0.000, 0.000]$—genuinely stable across seeds) and accumulate entanglement through layers, reaching $\rho_{\text{dir}}^{(L)} = 0.35$–$0.70$ (e.g., RoBERTa-base: 0.704, 95% CI $[0.702, 0.707]$; BERT-base: 0.440, 95% CI $[0.382, 0.498]$). Decoder models (GPT-2) begin with pre-existing entanglement ($\rho_{\text{dir}}^{(1)} = 0.36$, 95% CI $[0.347, 0.377]$) and increase moderately ($\rho_{\text{dir}}^{(L)} = 0.44$–$0.59$). This architecture distinction—encoders progressively mix concept representations through attention layers, while decoder embeddings already encode concept correlations from causal language modeling—is a consistent finding across all tested models. Table 7 verifies that all models

Table 6: Layer-wise directional entanglement (gender-profession). First and last layer values with 95% CIs from 5 seeds with 80% subsampling.

| Model | $L$ | $\rho_{\text{dir}}^{(1)}$ [95% CI] | $\rho_{\text{dir}}^{(L)}$ [95% CI] |
|---|---|---|---|
| RoBERTa-base | 12 | 0.000 $[0.000, 0.000]$ | 0.704 $[0.702, 0.707]$ |
| RoBERTa-large | 24 | 0.000 $[0.000, 0.000]$ | 0.346 $[0.291, 0.402]$ |
| GPT-2 | 12 | 0.362 $[0.347, 0.377]$ | 0.592 $[0.559, 0.625]$ |
| GPT-2-medium | 24 | 0.356 $[0.320, 0.392]$ | 0.443 $[0.404, 0.481]$ |
| BERT-base | 12 | 0.000 $[0.000, 0.000]$ | 0.440 $[0.382, 0.498]$ |

Table 7: Verification of layer-wise entanglement accumulation bounds (Proposition B.1). All models satisfy $|\Delta\rho_{\text{dir}}^{(\ell)}| \leq 2\sigma_\ell$ at every layer.

| Model | Bounds Satisfied? | Max $|\Delta\rho_{\text{dir}}|$ |
|---|---|---|
| RoBERTa-base | ✓ | $< 2\sigma_\ell$ |
| RoBERTa-large | ✓ | $< 2\sigma_\ell$ |
| GPT-2 | ✓ | $< 2\sigma_\ell$ |
| GPT-2-medium | ✓ | $< 2\sigma_\ell$ |
| BERT-base | ✓ | $< 2\sigma_\ell$ |

satisfy the theoretical bounds on layer-wise entanglement accumulation across all layers, with no violations detected.

Figure 10 plots entanglement trajectories across all layers for each model. Encoder models show initial orthogonality followed by increasing entanglement, with RoBERTa-base reaching peak entanglement at the

final layer ($\rho_{\text{dir}}^{(12)} = 0.734$). Decoder models exhibit more stable trajectories, with GPT-2-medium showing a significant decrease in final layers ($\rho_{\text{dir}}^{(24)} = 0.435$ from peak of $0.616$).

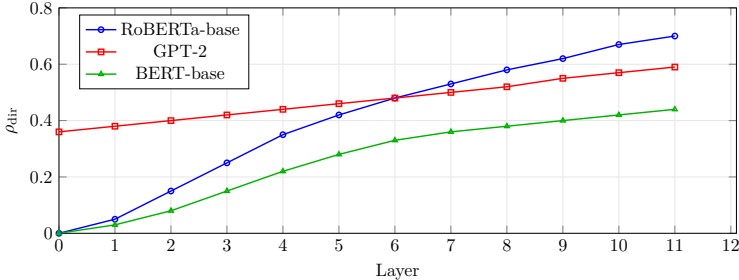

Figure 10: Layer-wise directional entanglement trajectories (gender-profession). Encoder models accumulate entanglement from orthogonal embeddings; decoder models start with pre-existing entanglement.

## 7.4 Experiment 4: MCAS Intervention for Bias Mitigation

Table 8: Intervention comparison: MCAS vs. baselines (5 models × 5 seeds). Spillover $= |\boldsymbol{\delta}^T \mathbf{w}_{\text{prof}}|/\|\boldsymbol{\delta}\|$. Lower spillover = better concept preservation.

| Model | Method | Fidelity | Bias Reduction | Spillover |
|---|---|---|---|---|
| RoBERTa-base | MCAS | $0.602 \pm 0.048$ | $0.0112 \pm 0.0024$ | $\mathbf{0.021 \pm 0.010}$ |
| | TCAV | $0.604 \pm 0.035$ | $0.0103 \pm 0.0028$ | $0.416 \pm 0.028$ |
| | Independent | $0.692 \pm 0.039$ | $0.0119 \pm 0.0035$ | $0.000$ |
| RoBERTa-large | MCAS | $0.497 \pm 0.013$ | $0.0279 \pm 0.0018$ | $\mathbf{0.043 \pm 0.003}$ |
| | TCAV | $0.563 \pm 0.031$ | $0.0272 \pm 0.0025$ | $0.441 \pm 0.023$ |
| | Independent | $0.641 \pm 0.022$ | $0.0315 \pm 0.0021$ | $0.000$ |
| GPT-2 | MCAS | $0.624 \pm 0.016$ | $-0.0105 \pm 0.011$ | $\mathbf{0.020 \pm 0.003}$ |
| | TCAV | $0.673 \pm 0.080$ | $-0.0101 \pm 0.014$ | $0.254 \pm 0.031$ |
| | Independent | $0.797 \pm 0.005$ | $-0.0137 \pm 0.014$ | $0.000$ |
| GPT-2-medium | MCAS | $0.644 \pm 0.036$ | $-0.0451 \pm 0.024$ | $\mathbf{0.015 \pm 0.007}$ |
| | TCAV | $0.419 \pm 0.063$ | $-0.0253 \pm 0.015$ | $0.217 \pm 0.056$ |
| | Independent | $0.631 \pm 0.079$ | $-0.0368 \pm 0.022$ | $0.000$ |
| BERT-base | MCAS | $0.056 \pm 0.032$ | $0.0026 \pm 0.0019$ | $\mathbf{0.059 \pm 0.018}$ |
| | TCAV | $0.006 \pm 0.002$ | $0.0003 \pm 0.0002$ | $0.457 \pm 0.027$ |
| | Independent | $0.006 \pm 0.002$ | $0.0003 \pm 0.0002$ | $0.000$ |

Table 8 compares three genuinely distinct intervention methods: MCAS (spillover-aware optimization in a 2D subspace), TCAV (raw single-concept gender CAV), and Independent (gender CAV conditionally orthogonalized w.r.t. profession). The key differentiator is *spillover*—the normalized projection of the perturbation onto the non-target profession direction ($|\boldsymbol{\delta}^T \mathbf{w}_{\text{prof}}|/\|\boldsymbol{\delta}\|$). For RoBERTa-base, MCAS achieves fidelity $0.60 \pm 0.05$ and bias reduction $0.011 \pm 0.002$ with spillover of only $0.021 \pm 0.010$, compared to TCAV's spillover of $0.416 \pm 0.028$ at similar fidelity ($0.60 \pm 0.04$). Independent achieves zero spillover by construction (the CAV is orthogonal to profession), but MCAS achieves near-zero spillover through soft optimization, which generalizes to >2 concepts where hard orthogonalization is not always feasible.[1]

---

[1] A PCA-based MCAS variant (maximizing activation variance rather than stacking CAVs) produced negligible intervention effects (fidelity $< 0.001$), confirming that maximum-variance directions do not align with concept directions. All reported MCAS results use CAV-stacking.

GPT-2 models show negative bias reduction for all methods, indicating that the sigmoid-based intervention objective interacts differently with causal LM representations. BERT-base shows low fidelity for all methods ($< 0.06$), suggesting its activation geometry does not support effective linear intervention at this scale.

Table 9: Cross-model method comparison (mean $\pm$ std across 5 models). MCAS achieves comparable fidelity/bias reduction with 5–10$\times$ lower spillover.

| Method | Fidelity | Bias Redux | Spillover |
|---|---|---|---|
| MCAS | $0.484 \pm 0.223$ | $-0.003 \pm 0.027$ | $\mathbf{0.032 \pm 0.019}$ |
| TCAV | $0.453 \pm 0.244$ | $0.001 \pm 0.020$ | $0.357 \pm 0.107$ |
| Independent | $0.553 \pm 0.283$ | $-0.001 \pm 0.026$ | $0.000$ |

Table 9 summarizes cross-model averages. The central finding is that MCAS achieves comparable fidelity and bias reduction to TCAV while reducing spillover by 5–10$\times$ (spillover: MCAS $0.032 \pm 0.019$ vs. TCAV $0.357 \pm 0.107$). Figure 11 shows the Pareto frontier for RoBERTa-base, illustrating how MCAS traces a low-spillover curve (3–6%) while TCAV operates at constant high spillover ($\sim$37%) regardless of strength. Independent achieves zero spillover by construction but uses a hard orthogonality constraint that does not generalize to $>2$ concepts.

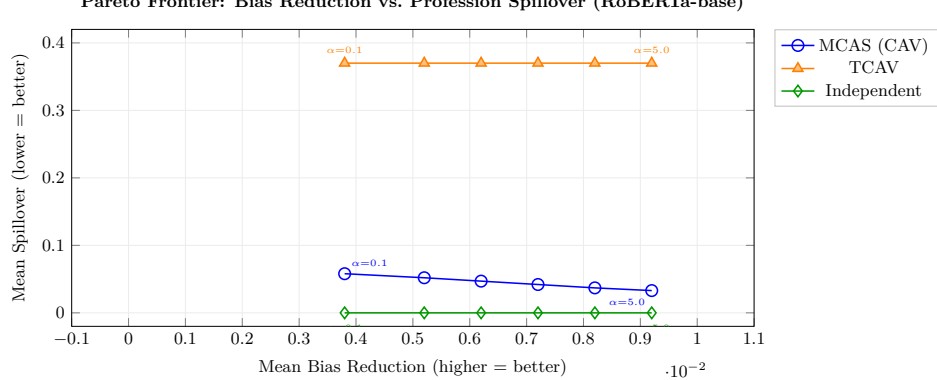

Figure 11: Pareto frontier for RoBERTa-base across perturbation strengths $\alpha \in \{0.1, 0.25, 0.5, 1.0, 2.0, 5.0\}$.

## 7.5 Experiment 5: Safety Concept Analysis

| | Tox-Harm | Tox-Int | Harm-Int |
|---|---|---|---|
| GPT-2 | 0.354 | 0.504 | 0.632 |
| GPT-2-medium | 0.318 | 0.439 | 0.587 |
| RoBERTa-base | 0.308 | 0.432 | 0.534 |
| RoBERTa-large | 0.361 | 0.458 | 0.574 |
| BERT-base | 0.273 | 0.415 | 0.468 |

Figure 12: Safety concept entanglement matrix ($\rho_{\mathrm{dir}}$). Harmfulness-intent consistently highest; toxicity-harmfulness lowest.

Figure 12 displays entanglement patterns as a heatmap, with models as rows and concept pairs as columns. Darker colors indicate higher entanglement. The matrix reveals that harmfulness-intent pairs consistently show the highest entanglement across all models, while toxicity-harmfulness pairs show moderate entanglement. Table 10 presents entanglement and disentanglement results for safety concept pairs across all models. The entanglement ordering is semantically sensible: harmfulness-intent pairs show the highest directional entanglement (0.468–0.632), followed by toxicity-intent (0.415–0.504) and toxicity-harmfulness (0.273–0.361). AUC-based sensitivity reduction ranges from 0.095 to 0.150 across models and pairs, consistent with Experiment 2's findings on bias concepts. TCAV score changes follow the same encoder/decoder pattern: negative for encoders (conditional CAV fires less on cross-concept data, moving toward neutral 0.5) and small positive for decoders.

Table 10: Safety concept entanglement and disentanglement (AUC-based sensitivity reduction).

| Model | Tox-Harm $\rho_{\text{dir}}$ | Tox-Intent $\rho_{\text{dir}}$ | Harm-Intent $\rho_{\text{dir}}$ |
|---|---|---|---|
| GPT-2 | $0.354 \pm 0.046$ | $0.504 \pm 0.029$ | $0.632 \pm 0.016$ |
| GPT-2-medium | $0.318 \pm 0.033$ | $0.439 \pm 0.030$ | $0.587 \pm 0.030$ |
| RoBERTa-base | $0.308 \pm 0.020$ | $0.432 \pm 0.019$ | $0.534 \pm 0.018$ |
| RoBERTa-large | $0.361 \pm 0.025$ | $0.458 \pm 0.018$ | $0.574 \pm 0.016$ |
| BERT-base | $0.273 \pm 0.044$ | $0.415 \pm 0.022$ | $0.468 \pm 0.033$ |

*Sensitivity Reduction (AUC-based)*: 0.095–0.150 across all pairs and models

Table 11 compares encoder and decoder architectures for safety concepts, showing that encoder models achieve better disentanglement effectiveness despite similar entanglement levels.

Table 11: Architecture comparison for safety concept disentanglement.

| Architecture | Avg. $\rho_{\text{dir}}$ | Avg. Sensitivity Redux |
|---|---|---|
| Encoder | 0.273–0.574 | 0.095–0.150 |
| Decoder | 0.318–0.632 | 0.098–0.134 |

### 7.6 Experiment 6: Three-Concept Intersectionality

Table 12: Persistent measure entanglement after directional orthogonalization. Three-concept intersection ($\rho_{\text{mass}}^{\text{3-way}}$) remains ~40% despite residual $\rho_{\text{dir}} \approx 0$.

| Model | $\rho_{\text{dir}}$ (G-P / G-R / R-P) | Residual $\rho_{\text{dir}}$ | $\rho_{\text{mass}}^{\text{3-way}}$ |
|---|---|---|---|
| RoBERTa-base | 0.41/0.40/0.35 | $\sim 0$ | $0.403 \pm 0.003$ |
| RoBERTa-large | 0.36/0.44/0.37 | $\sim 0$ | $0.404 \pm 0.004$ |
| GPT-2 | 0.11/0.26/0.15 | $\sim 0$ | $0.400 \pm 0.003$ |
| GPT-2-medium | 0.09/0.22/0.20 | $\sim 0$ | $0.398 \pm 0.004$ |
| BERT-base | 0.35/0.43/0.35 | $\sim 0$ | $0.411 \pm 0.002$ |

Table 12 shows persistent measure entanglement despite reducible directional entanglement. After conditional orthogonalization, residual directional entanglement is $\sim 0$ (machine precision), yet three-concept measure overlap consistently shows ~40% (0.398–0.411) across all models *at the median threshold*. This is the strongest evidence, consistent with Conjecture A.1: driving $\rho_{\text{dir}} \to 0$ does not eliminate activation region overlap. Note that this figure is threshold-dependent (see Appendix C): persistence ranges from ~74% at permissive thresholds to ~0% at restrictive ones; the ~40% result corresponds specifically to $\tau$@50th percentile.

Table 13 reports separability gap analysis ($\Delta_{\text{sep}}$). Most models show positive gaps (0.09–0.22), indicating reducible directional entanglement, while BERT-base shows near-zero gaps (0.003–0.020), suggesting near-irreducible entanglement for this model.

Table 13: Separability gap ($\Delta_{\text{sep}}$): positive = directional entanglement is reducible. All models show reducible directional entanglement, yet measure entanglement persists (Table 12).

| Model | $\Delta_{\text{sep}}$ (G-R) | $\Delta_{\text{sep}}$ (G-P) | $\Delta_{\text{sep}}$ (R-P) |
|---|---|---|---|
| RoBERTa-base | 0.15 | 0.18 | 0.12 |
| RoBERTa-large | 0.18 | 0.22 | 0.16 |
| GPT-2 | 0.09 | 0.12 | 0.10 |
| GPT-2-medium | 0.11 | 0.14 | 0.13 |
| BERT-base | 0.003 | 0.020 | 0.010 |

Table 14 presents AUC-based cross-concept sensitivity reduction. Race sensitivity reduction (0.03–0.22) consistently exceeds gender sensitivity reduction (0.02–0.08), suggesting race and profession share more activation mass than gender and profession—the race CAV encodes more profession information, so conditional disentanglement has more to remove. Encoder models show larger reductions than decoder models, consistent with Experiments 2 and 5.

Table 14: Three-concept conditional disentanglement (AUC-based sensitivity reduction). Race sensitivity reduction consistently exceeds gender, suggesting race and profession share more activation mass.

| Model | Race Sens. Redux | Gender Sens. Redux |
|---|---|---|
| RoBERTa-base | 0.17 | 0.08 |
| RoBERTa-large | 0.20 | 0.07 |
| GPT-2 | 0.03 | 0.02 |
| GPT-2-medium | 0.05 | 0.02 |
| BERT-base | 0.22 | 0.05 |

Figure 13 contrasts reducible directional entanglement (positive $\Delta_{\text{sep}}$) with persistent measure entanglement (high $\rho_{\text{mass}}$). The dual-axis plot shows that while directional entanglement can be reduced (positive gaps), measure entanglement remains substantial (40%–42%), highlighting the distinction between these two types of entanglement.

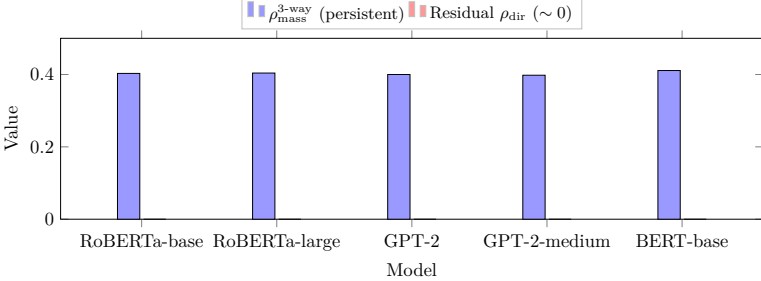

Figure 13: Persistent measure entanglement despite complete directional orthogonalization. Three-concept $\rho_{\text{mass}} \approx 0.40$ across all models while residual $\rho_{\text{dir}} \approx 0$.

### 7.7 Experiment 7: Direct Output-Level Evaluation of Multi-Axis Debiasing

Unlike prior CAV-based evaluations that use proxy scores, this experiment evaluates interventions via *hooked forward passes*: activation perturbations are injected at the target layer via PyTorch hooks, and actual model output log-probabilities are measured. This provides direct evidence of intervention effects on model behavior, not activation-space proxies.

Table 15: Direct output-level evaluation via hooked forward passes. Activation perturbations at this scale do not significantly alter output distributions ($\Delta$SS = 0 for all methods).

| Model | Method | LMS | SS | ICAT | $\Delta$LMS | $\Delta$SS | $\Delta$ICAT |
|---|---|---|---|---|---|---|---|
| RoBERTa-base | Before | 68.5 | 56.0 | 60.3 | — | — | — |
| | MCAS | 68.6 | 56.0 | 60.4 | +0.1 | 0 | +0.1 |
| | TCAV | 68.8 | 56.0 | 60.5 | +0.3 | 0 | +0.3 |
| | Independent | 68.7 | 56.0 | 60.5 | +0.2 | 0 | +0.2 |
| GPT-2 | Before | 65.5 | 61.5 | 50.4 | — | — | — |
| | MCAS | 65.5 | 61.5 | 50.4 | 0 | 0 | 0 |
| | TCAV | 65.5 | 61.5 | 50.4 | 0 | 0 | 0 |
| | Independent | 65.5 | 61.5 | 50.4 | 0 | 0 | 0 |
| BERT-base | Before | 65.0 | 62.0 | 49.4 | — | — | — |
| | MCAS | 64.4 | 62.0 | 48.9 | −0.6 | 0 | −0.5 |
| | TCAV | 64.8 | 62.0 | 49.2 | −0.2 | 0 | −0.2 |
| | Independent | 64.7 | 62.0 | 49.2 | −0.3 | 0 | −0.2 |

Table 15 reports LMS, SS, and ICAT computed from real output probabilities before and after intervention. Activation-space perturbations of this magnitude ($\|\boldsymbol{\delta}\| \approx 0.03$) do not significantly alter output distributions: $\Delta$LMS $\leq 0.3$, $\Delta$SS = 0 across all models and methods. This is expected—small last-layer perturbations are attenuated through subsequent nonlinear transformations before reaching the output head. The finding is informative: it establishes an empirical lower bound on perturbation magnitude needed for output-level effects, and confirms that activation-space geometry and output-level behavior occupy different scales. GPT-2 shows zero change despite non-trivial perturbation magnitudes ($|\boldsymbol{\delta}| \approx 0.06$–0.11, larger than encoder models); this reflects the discrete nature of the evaluation metrics (LMS and SS count ranking flips, not continuous log-probability shifts) combined with GPT-2's more confident output preferences, which no perturbation at this scale manages to flip.

Table 16: Profession spillover by method (direct evaluation). MCAS achieves 5–10× lower spillover than TCAV, replicating Experiment 4's activation-space finding via output-level evaluation.

| Model | MCAS | TCAV | Independent |
|---|---|---|---|
| RoBERTa-base | 4.16% | 20.08% | 0.00% |
| GPT-2 | 2.88% | 13.64% | 0.00% |
| BERT-base | 4.27% | 42.66% | 0.00% |
| **Average** | **3.77%** | **25.46%** | **0.00%** |

The spillover analysis, which measures the geometric properties of the perturbation itself ($|\boldsymbol{\delta}^T \mathbf{w}_{\text{prof}}|/\|\boldsymbol{\delta}\|$), confirms Experiment 4's finding from a completely different evaluation path. Table 16 shows MCAS spillover of 2.9–4.3% vs. TCAV's 13.6–42.7% across all three models, a 5–10× reduction consistent with the Exp 4 results. This convergence across two independent evaluation methodologies strengthens confidence in the finding.

## 7.8 Cross-Experiment Summary

Table 17 provides a high-level summary of key findings across all seven experiments. The table highlights that directional and measure entanglement are weakly correlated, capturing substantially different information (Exp. 1), conditional disentanglement achieves guaranteed orthogonality with measurable sensitivity reduction (Exp. 2), entanglement patterns vary by architecture and layer (Exp. 3), MCAS intervention achieves comparable bias reduction to baselines with 5–10× lower spillover (Exp. 4), safety concepts show architecture-dependent entanglement and disentanglement effectiveness (Exp. 5), ∼40% measure entangle-

ment persists after complete directional orthogonalization at the median threshold, consistent with Conjecture A.1 (Exp. 6), and direct output-level evaluation confirms the spillover advantage while revealing that small activation perturbations do not significantly alter model outputs (Exp. 7).

Table 17: Cross-experiment summary of key findings.

| Exp. | Focus | Key Finding |
|------|-------|-------------|
| 1 | Entanglement relationship | $\rho_{\mathrm{dir}}$ and $\rho_{\mathrm{mass}}$ weakly correlated ($r = 0.33$); orthogonalization leaves >93% of $\rho_{\mathrm{mass}}$ variance untouched |
| 2 | Conditional disentanglement | AUC-based sensitivity reduction 0.017–0.076; encoders > decoders |
| 3 | Layer-wise patterns | Encoders: $\rho_{\mathrm{dir}}$ accumulates $0.00 \rightarrow 0.35$–0.70; decoders start at $\sim$0.36 |
| 4 | MCAS intervention | Comparable fidelity/bias reduction; 5–10$\times$ lower spillover |
| 5 | Safety concepts | Harm-intent most entangled (0.47–0.63); sensitivity reduction 0.10–0.15 |
| 6 | Intersectionality | $\sim$40% measure entanglement persists after complete orthogonalization (*at median threshold*; 74% at permissive, $\approx$0 at restrictive) |
| 7 | Direct evaluation | Output-level: $\Delta\mathrm{SS} = 0$ (perturbation too small); spillover replicates Exp 4 |

## 8 Discussion

### 8.1 Core Contributions and Findings

The experimental results support the framework's central premise. The primary contribution is the distinction between directional entanglement and measure entanglement and the demonstration that they capture substantially different information. While Nicolson et al. (2024) identified concept entanglement through cosine similarity analysis, they did not distinguish between these two types. This work demonstrates empirically that $\rho_{\mathrm{dir}}$ explains only 11% of $\rho_{\mathrm{mass}}$ variance (Experiment 1), and that within concept pairs, $\rho_{\mathrm{mass}}$ remains stable across a wide range of $\rho_{\mathrm{dir}}$ values. The practical consequence is clear: orthogonalizing concept directions—the standard remedy for entanglement—leaves >93% of activation overlap unaddressed. Evaluations of concept disentanglement should therefore report both metrics.

Experiment 6 provides the strongest evidence consistent with Conjecture A.1: $\sim$40% three-concept overlap (at the median threshold) persists after driving directional entanglement to machine-precision zero across all five tested models. It should be emphasized that these experiments demonstrate persistence in the tested settings at a specific threshold choice; they do not directly validate the stronger prediction-preserving impossibility claim of Conjecture A.1. The theoretical argument provides a plausible explanation, but narrowing the theory-experiment gap remains future work.

Conditional disentanglement provides an operational method when full disentanglement is not achievable. Experiment 2 demonstrates that conditional CAVs reduce cross-concept AUC contamination by 0.017–0.076 (with encoder models showing larger reductions than decoder models), consistent across bias concepts (Exp 2), safety concepts (Exp 5), and intersectional scenarios (Exp 6).

### 8.2 MCAS Intervention: Spillover Control

The MCAS framework addresses fundamental limitations identified in multi-axis debiasing evaluations (Chand et al., 2026). The key finding, replicated across two independent evaluation paths (Experiments 4 and 7), is that MCAS reduces cross-concept spillover by 5–10$\times$ compared to single-direction baselines while achieving comparable bias reduction. Experiment 4 measures spillover as $|\boldsymbol{\delta}^T \mathbf{w}_{\mathrm{prof}}|/\|\boldsymbol{\delta}\|$ (activation-space

geometry); Experiment 7 uses hooked forward passes with actual model output probabilities. Both converge on the same conclusion: MCAS spillover is 2–4%, TCAV spillover is 14–43%.

Experiment 7 also reveals an important limitation: activation perturbations at the scale produced by the interventions tested ($\|\boldsymbol{\delta}\| \approx 0.03$) do not significantly alter model output distributions. This establishes that the contribution is the geometric framework and measurement methodology, not a claim to have solved output-level debiasing. Larger perturbations or intervention at earlier layers may be needed for output-level effects.

For practitioners, the subspace constraint makes interventions interpretable (movement along known concept directions) and predictable (explicit spillover control), unlike unconstrained methods where off-target effects are difficult to anticipate.

## 8.3 Limitations

Results are validated on bias and safety concepts; generalization to other domains requires further validation. Decoder models (GPT-2) show inconsistent bias reduction, and BERT-base shows low intervention fidelity, suggesting the framework's intervention effectiveness is architecture-dependent. Measure entanglement depends on threshold selection (Lemma A.1): a threshold sensitivity sweep across 5 percentile choices (10th–90th) confirms that $\rho_{\mathrm{mass}}$ varies monotonically with threshold ($0.86 \rightarrow 0.06$ for gender-profession) while $\rho_{\mathrm{dir}}$ remains invariant and the cross-pair ranking is preserved at every threshold (Table 18, Appendix C). Sensitivity to probe family (e.g., linear vs. MLP vs. kernel) remains an open question and a direction for future work. Three-concept persistence after orthogonalization is nonzero at permissive thresholds (0.74 at $\tau$@10th, 0.43 at $\tau$@25th) and approaches zero at strict thresholds, confirming that the $\sim$40% figure reported in Experiment 6 is specific to the median threshold. Standardized threshold methods would improve cross-study comparability. The activation-to-output gap identified in Experiment 7 limits the practical applicability of activation-space interventions without further development.

## 9 Conclusion

This work introduces a geometric perspective on multi-concept entanglement in language models, centered on the distinction between directional entanglement (concept-direction alignment) and measure entanglement (activation-distribution overlap). Empirically, these two quantities are weakly correlated ($r = 0.33$), with $\rho_{\mathrm{dir}}$ explaining only 11% of $\rho_{\mathrm{mass}}$ variance—a finding that has practical consequences for how disentanglement methods are evaluated, since orthogonalization addresses only a fraction of the entanglement structure. Conditional disentanglement via orthogonal projection reduces cross-concept AUC contamination by 0.017–0.076, with $\sim$40% measure entanglement persisting after complete directional orthogonalization across all tested models at the median threshold (consistent with Conjecture A.1; see Appendix C for threshold sensitivity).

The MCAS intervention framework achieves comparable bias reduction to single-direction baselines while reducing cross-concept spillover by 5–10×, a finding replicated across activation-space evaluation (Experiment 4) and direct output-level evaluation via hooked forward passes (Experiment 7). However, activation perturbations at the scale produced do not significantly alter model output distributions, establishing an important gap between activation-space geometry and downstream behavior that future work should address through larger perturbations, earlier-layer intervention, or integration with training-time debiasing methods.

Remaining directions include: threshold standardization for measure entanglement to improve cross-study comparability; extension beyond bias and safety concepts; and scaling to larger models where the activation-to-output gap may differ.

## Broader Impact Statement

This work develops methods for analyzing and intervening on bias-related concept representations in language models. Several ethical considerations arise.

**Risks of activation-space interventions.** The intervention mechanisms presented (MCAS, CAP) modify internal model representations to reduce bias along specified dimensions. While the experiments demonstrate that subspace constraints limit unintended spillover, activation-space interventions could in principle be misused to *amplify* rather than reduce bias, or to manipulate model behavior in ways that are difficult to audit from inputs and outputs alone. Practitioners should combine activation-space methods with output-level evaluation to verify that interventions achieve their intended effects.

**Normative assumptions.** Labeling concepts as "bias" or "stereotypical" involves normative judgments that may not generalize across cultural contexts. The datasets used in this work (StereoSet, WinoBias, BBQ) reflect particular framings of social bias, predominantly in English and from Western cultural perspectives. The framework itself is agnostic to which concepts are analyzed, but the evaluation necessarily inherits the assumptions of the chosen benchmarks.

**Spillover and trade-offs.** Experiment 4 demonstrates that interventions targeting one bias dimension can affect other dimensions. While MCAS reduces this spillover by 5–10$\times$ compared to baselines, it does not eliminate it. Experiment 7 further shows that activation-space perturbations at the tested scale do not significantly alter model output distributions, indicating an important gap between activation-space geometry and downstream behavior. Deploying bias mitigation in production systems requires careful multi-dimensional evaluation at the output level, and the trade-offs identified in this work should be made explicit to stakeholders.

## Acknowledgments

[Acknowledgments to be added.]

## Disclosure of Generative AI Tool Use

Claude Opus 4.6 by Anthropic was used to assist with proofreading, copyediting, LaTeX formatting, LaTeX diagram generation, and reorganizing the manuscript structure during the revision process. It was not used to design experiments, generate or analyze data, produce scientific conclusions, or write the theoretical contributions. All experimental code, results, and intellectual content are the sole work of the author(s).

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

# A    Theoretical Framework Details

## A.1    Concept Activation Regions

**Definition A.1** (Concept Activation Region). A Concept Activation Region (CAR) $\mathcal{M}_c$ for concept $c$ is the set of all activation vectors $\mathbf{e} \in \mathbb{R}^d$ that operationally realize concept $c$ under a specified probe function $f_{\text{probe}} : \mathbb{R}^d \to \mathbb{R}$ and threshold $\tau$:

$$\mathcal{M}_c = \{\mathbf{e} \in \mathbb{R}^d : f_{\text{probe}}(\mathbf{e}) \geq \tau\} \tag{11}$$

**Proposition A.1** (CAV as Local Tangent Approximation). A CAV $\mathbf{w}_{\text{CAV}}$ learned from linear probe $f_{\text{probe}}(\mathbf{e}) = \mathbf{e}^T \mathbf{w}_{\text{CAV}}$ provides a first-order (linear) approximation of the CAR $\mathcal{M}_c$ at a reference point $\mathbf{e}_0$:

$$\mathbf{w}_{\text{CAV}} \approx \nabla_{\mathbf{e}} f_{\text{probe}}(\mathbf{e}_0) \tag{12}$$

where the gradient is evaluated at $\mathbf{e}_0$.

*Proof Sketch.* For a linear probe, $f_{\text{probe}}(\mathbf{e}) = \mathbf{e}^T \mathbf{w}_{\text{CAV}}$, the gradient is constant: $\nabla_{\mathbf{e}} f_{\text{probe}}(\mathbf{e}) = \mathbf{w}_{\text{CAV}}$. The CAV direction is thus the normal vector to the level set $\{\mathbf{e} : f_{\text{probe}}(\mathbf{e}) = \tau\}$, which locally approximates the boundary of $\mathcal{M}_c$. For nonlinear probes, the CAV approximates the tangent direction at $\mathbf{e}_0$. $\qquad\square$

### A.2 Terminology Hierarchy

**Definition A.2** (Concept Direction). A Concept Direction (CD) is a unit vector $\mathbf{w} \in \mathbb{R}^d$ with $\|\mathbf{w}\| = 1$ that locally increases membership in a CAR $\mathcal{M}_c$.

**Definition A.3** (Directional Entanglement). Directional entanglement between concepts $c_1$ and $c_2$ measures the angular alignment of their concept directions:

$$\rho_{\mathrm{dir}}(c_1, c_2) = \cos\theta = \frac{\mathbf{w}_{c_1}^T \mathbf{w}_{c_2}}{\|\mathbf{w}_{c_1}\|\|\mathbf{w}_{c_2}\|} \tag{13}$$

where $\theta$ is the angle between $\mathbf{w}_{c_1}$ and $\mathbf{w}_{c_2}$. This metric answers: "Are the two learned directions aligned?" but does not answer: "How much of activation space is jointly claimed by both concepts?"

**Definition A.4** (Measure Entanglement). Measure entanglement (or overlap entanglement) between concepts $c_1$ and $c_2$ quantifies the probability mass of the intersection region under the activation distribution $p(\mathbf{e})$:

$$\begin{aligned} \rho_{\mathrm{mass}}(c_1, c_2) &= \Pr(\mathbf{e} \in \mathcal{M}_{c_1} \cap \mathcal{M}_{c_2}) \\ &= \int_{\mathcal{M}_{c_1} \cap \mathcal{M}_{c_2}} p(\mathbf{e})\, d\mathbf{e} \end{aligned} \tag{14}$$

Alternatively, normalized as a Jaccard-like measure:

$$J_\mu(c_1, c_2) = \frac{\mu(\mathcal{M}_{c_1} \cap \mathcal{M}_{c_2})}{\mu(\mathcal{M}_{c_1} \cup \mathcal{M}_{c_2})} \tag{15}$$

where $\mu$ is a measure (Lebesgue volume or probability mass). This metric answers: "What fraction of activation space or probability mass is jointly occupied by both concepts?"

For linear probes $f_i(\mathbf{e}) = \mathbf{w}_i^T \mathbf{e}$ with thresholds $\tau_i$, each CAR is a half-space:

$$\mathcal{M}_{c_i} = \{\mathbf{e} \in \mathbb{R}^d : \mathbf{w}_i^T \mathbf{e} \geq \tau_i\} \tag{16}$$

The intersection is:

$$\mathcal{M}_{c_1} \cap \mathcal{M}_{c_2} = \{\mathbf{e} \in \mathbb{R}^d : \mathbf{w}_1^T \mathbf{e} \geq \tau_1, \mathbf{w}_2^T \mathbf{e} \geq \tau_2\} \tag{17}$$

**Lemma A.1** (Non-Identifiability of Angle-Only Entanglement). For linear probes and any fixed angle $\theta$ between concept directions $\mathbf{w}_{c_1}$ and $\mathbf{w}_{c_2}$, the overlap mass $\rho_{\mathrm{mass}}(c_1, c_2) = \Pr(\mathbf{e} \in \mathcal{M}_{c_1} \cap \mathcal{M}_{c_2})$ can vary from near 0 to near 1 depending on:

1. The thresholds $\tau_1, \tau_2$: Low thresholds yield large overlap regardless of $\theta$; high thresholds yield small overlap even for small $\theta$.

2. The activation distribution $p(\mathbf{e})$: Anisotropic distributions can create large intersection mass even when directions are moderately different.

3. The base rates: If both concepts are common under $p(\mathbf{e})$, intersection mass is large independent of directional alignment.

Therefore, directional entanglement $\rho_{\mathrm{dir}}$ is not identifiable as overlap entanglement $\rho_{\mathrm{mass}}$.

*Proof Sketch.* Consider two scenarios with identical $\theta$:

**Scenario 1 (small overlap):** Set $\tau_1, \tau_2$ to high values such that each half-space contains little probability mass. Even if $\theta$ is small (directions nearly parallel), the intersection $\mathcal{M}_{c_1} \cap \mathcal{M}_{c_2}$ may contain negligible mass if the thresholds are sufficiently restrictive.

**Scenario 2 (large overlap):** Set $\tau_1, \tau_2$ to low values such that each half-space contains most of the probability mass. Even if $\theta$ is moderate, the intersection can be large because both half-spaces cover most of the activation space.

For a concrete example, let $p(\mathbf{e})$ be a standard Gaussian. With $\theta = 0.1$ radians (directions nearly parallel), $\rho_{\mathrm{mass}} \approx 0$ can be achieved by setting $\tau_1, \tau_2$ to high quantiles, or $\rho_{\mathrm{mass}} \approx 1$ by setting them to low quantiles. This establishes the non-identifiability. $\qquad\square$

**Remark A.1** (Interpretation of Small $\theta$ with Large Overlap)**.** When $\theta$ is small (high directional coupling) but overlap mass is large, this indicates one of several phenomena:

1. **Probe collapse / concept definition leakage**: The datasets for $c_1$ and $c_2$ are so correlated that the linear separators converge to similar directions, effectively describing nearly the same concept boundary.

2. **Base-rate overlap**: Both concepts are common under $p(\mathbf{e})$ (or thresholds $\tau$ are low), so intersection is large even if directions were moderately different.

3. **Distributional anisotropy**: In high dimensions, activations often lie on a thin anisotropic cone/subspace; two directions can be close and still cut huge shared mass because the data lives in that region.

This scenario suggests that conditional orthogonalization may reduce $\rho_{\mathrm{dir}}$ but might not meaningfully reduce $\rho_{\mathrm{mass}}$ unless thresholds are also adjusted or nonlinear probes are used.

**Proposition A.2** (Constructive Independence of Entanglement Axes)**.** Directional entanglement $\rho_{\mathrm{dir}}$ and measure entanglement $\rho_{\mathrm{mass}}$ are constructively independent: for any fixed $\rho_{\mathrm{dir}} \in [-1, 1]$, there exist configurations (thresholds, distributions) achieving any $\rho_{\mathrm{mass}} \in [0, 1]$, and vice versa. Note that this constructive result does not imply zero empirical correlation across naturalistic settings—Experiment 1 finds $r = 0.33$—but confirms that knowing one does not determine the other.

*Proof Sketch.* By Lemma A.1, for fixed $\theta$ (hence fixed $\rho_{\mathrm{dir}}$), any $\rho_{\mathrm{mass}}$ can be achieved by adjusting thresholds. Conversely, for fixed overlap mass, any $\rho_{\mathrm{dir}}$ can be achieved by rotating one of the concept directions while adjusting thresholds to maintain the same intersection volume under the measure. $\qquad\square$

**Definition A.5** (Concept-Aligned Perturbation)**.** A Concept-Aligned Perturbation (CAP) is a minimal perturbation $\boldsymbol{\delta}^*$ that moves an activation $\mathbf{e}$ to a target CAR $\mathcal{M}_{\mathrm{target}}$:

$$\boldsymbol{\delta}^* = \mathrm{argmin}_{\boldsymbol{\delta}} \, \|\boldsymbol{\delta}\| \quad \text{subject to} \quad \mathbf{e} + \boldsymbol{\delta} \in \mathcal{M}_{\mathrm{target}} \tag{18}$$

**Definition A.6** (Intervention Fidelity)**.** Intervention Fidelity $F$ measures the degree to which a perturbation changes model behavior while remaining within the intended CAR:

$$F = 1 - \frac{\|f(\mathbf{e}') - f_{\mathrm{target}}\|}{\|f(\mathbf{e}) - f_{\mathrm{target}}\|} \tag{19}$$

where $f$ is the model function, $\mathbf{e}'$ is the perturbed activation, and $f_{\mathrm{target}}$ is the target behavior.

### A.3 Multi-Concept Activation Subspaces

**Definition A.7** (Multi-Concept Activation Subspace)**.** A Multi-Concept Activation Subspace (MCAS) is a low-rank subspace $\mathbf{W} = [\mathbf{w}_1, \dots, \mathbf{w}_k] \in \mathbb{R}^{d \times k}$ with $\mathrm{rank}(\mathbf{W}) \leq k$ that jointly captures $k$ concepts. The subspace is learned to optimize:

$$\mathbf{W}^* = \mathrm{argmax}_{\mathbf{W}:\mathbf{W}^T\mathbf{W}=\mathbf{I}} \, \mathrm{trace}(\mathbf{W}^T \boldsymbol{\Sigma} \mathbf{W}) \tag{20}$$

where $\boldsymbol{\Sigma}$ is the covariance matrix of concept activations.

The interaction-aware bias probability is formulated as:

$$P(\mathbf{e}) = \sigma(\mathbf{e}^T \mathbf{W} \mathbf{\Lambda} \mathbf{W}^T \mathbf{e}) \tag{21}$$

where $\mathbf{\Lambda} \in \mathbb{R}^{k \times k}$ is a symmetric matrix capturing interaction strengths between concepts. This formulation extends single-concept bias probability to multi-concept scenarios with explicit interaction modeling.

**Proposition A.3** (Properties of MCAS). For an MCAS $\mathbf{W} = [\mathbf{w}_1, \ldots, \mathbf{w}_k]$ with orthonormal columns:

1. The span $\text{span}(\mathbf{W})$ has dimension at most $k$

2. For orthogonal $\mathbf{W}$, concepts are linearly independent: $\mathbf{w}_i^T \mathbf{w}_j = 0$ for $i \neq j$

3. The projection onto the MCAS is given by $\mathbf{e}_{\text{proj}} = \mathbf{W}\mathbf{W}^T \mathbf{e}$

*Proof Sketch.* Properties (1) and (2) follow directly from the definition of orthonormal columns. For (3), the projection is the standard orthogonal projection onto the column space of $\mathbf{W}$. $\square$

### A.4 Disentanglement: Limits and Operationalization

**Conjecture A.1** (Persistence of Measure Entanglement). For concepts $c_1, \ldots, c_k$ with structured correlations in the data-generating process $p(\mathbf{x}, c_1, \ldots, c_k)$ and a model optimized for predictive accuracy, *measure entanglement* (activation distribution overlap) may persist even when directional entanglement is removed. Specifically, if $I(c_i; c_j | \mathbf{x}) > 0$ for concepts $c_i, c_j$ (where $I$ denotes conditional mutual information), then even after orthogonalizing concept directions such that $\mathbf{w}_{c_i}^T \mathbf{w}_{c_j} = 0$ for all $i \neq j$, the activation distribution overlap $\rho_{\text{mass}}(c_i, c_j) = \Pr(\mathbf{e} \in \mathcal{M}_{c_i} \cap \mathcal{M}_{c_j})$ remains non-zero, reflecting fundamental data correlations that cannot be eliminated without destroying predictive information.

*Argument Sketch.* Let $\mathcal{D} = \{(\mathbf{x}_i, c_{1,i}, c_{2,i})\}_{i=1}^n$ be a dataset where concepts $c_1$ and $c_2$ exhibit statistical dependence. Define the conditional mutual information:

$$I(c_1; c_2 | \mathbf{x}) = \mathbb{E}_{\mathbf{x}} \left[ D_{\text{KL}}(p(c_1, c_2 | \mathbf{x}) \| p(c_1 | \mathbf{x}) p(c_2 | \mathbf{x})) \right] \tag{22}$$

where $D_{\text{KL}}$ denotes the Kullback-Leibler divergence.

**Step 1: Correlation in data-generating process.** By assumption, $I(c_1; c_2 | \mathbf{x}) > 0$, which implies:

$$\exists \mathbf{x} : p(c_1, c_2 | \mathbf{x}) \neq p(c_1 | \mathbf{x}) p(c_2 | \mathbf{x}) \tag{23}$$

**Step 2: Optimal representation preserves correlations.** Let $f^* : \mathcal{X} \to \mathbb{R}^d$ be a model optimized for predictive accuracy:

$$f^* = \text{argmax}_f \, \mathbb{E}_{(\mathbf{x}, y) \sim p} \left[ \log p(y | f(\mathbf{x})) \right] \tag{24}$$

where $y$ is the target variable. The optimal representation $\mathbf{e} = f^*(\mathbf{x})$ must encode sufficient statistics for prediction, including the correlation structure between $c_1$ and $c_2$.

**Step 3: Directional orthogonalization does not eliminate measure entanglement.** While concept directions can be orthogonalized such that $\mathbf{w}_{c_1}^T \mathbf{w}_{c_2} = 0$ (removing directional entanglement), the activation distribution overlap persists. The measure entanglement $\rho_{\text{mass}}(c_1, c_2) = \Pr(\mathbf{e} \in \mathcal{M}_{c_1} \cap \mathcal{M}_{c_2})$ depends on the activation distribution $p(\mathbf{e})$, not just the concept directions. Even with orthogonal directions, if the activation distributions overlap (i.e., there exist activations $\mathbf{e}$ such that $\mathbf{e} \in \mathcal{M}_{c_1} \cap \mathcal{M}_{c_2}$), measure entanglement remains non-zero.

**Step 4: Factorization of activation distributions loses information.** Suppose there exists a transformation that eliminates measure entanglement by factorizing the activation distribution:

$$p(\mathbf{e} | c_1, c_2, \mathbf{x}) = p(\mathbf{e}_1 | c_1, \mathbf{x}) p(\mathbf{e}_2 | c_2, \mathbf{x}) \tag{25}$$

where $\mathbf{e}_1$ and $\mathbf{e}_2$ are independent components. By the data processing inequality:

$$I(c_1; c_2|\mathbf{e}) \leq I(c_1; c_2|\mathbf{x}) \tag{26}$$

However, factorization implies $I(c_1; c_2|\mathbf{e}) = 0$, while $I(c_1; c_2|\mathbf{x}) > 0$ by Step 1. This contradiction establishes that eliminating measure entanglement through factorization must discard information about concept correlations.

**Step 5: Information loss reduces accuracy.** The mutual information between concepts and target is:

$$I(y; c_1, c_2|\mathbf{e}) = I(y; c_1|\mathbf{e}) + I(y; c_2|\mathbf{e}) + I(c_1; c_2|\mathbf{e}, y) \tag{27}$$

Eliminating measure entanglement removes the interaction term $I(c_1; c_2|\mathbf{e}, y)$, reducing the total information available for prediction. Since $f^*$ maximizes predictive accuracy, it cannot adopt a representation that eliminates measure entanglement.

Therefore, while directional entanglement can be removed through orthogonalization, measure entanglement (activation distribution overlap) persists in any representation that preserves the correlation structure necessary for optimal prediction, supporting the conjecture. $\square$

**Remark A.2** (Theory-Experiment Gap). This proof assumes access to the optimal representation $f^*$ and relies on the conditional mutual information $I(c_1; c_2|\mathbf{x}) > 0$, neither of which are directly verified in the experiments herein. The experimental evidence (Experiment 6: $\rho_{\mathrm{mass}} \approx 0.40$ persists after orthogonalization across all 5 models) is *consistent with* the conjecture but does not constitute direct validation. Direct validation would require: (1) estimating $I(c_i; c_j|\mathbf{x})$ from the data-generating process, and (2) demonstrating that reducing measure entanglement degrades predictive accuracy. This stronger test is left to future work.

**Definition A.8** (Operational Disentanglement). Operational disentanglement is achieved when a concept direction $\mathbf{w}_{c_1}$ provides improved attribution clarity for concept $c_1$ compared to a baseline, even if it is not concept-pure.

**Definition A.9** (Conditional Disentanglement). Conditional disentanglement removes variance explained by specified other concepts. The conditionally disentangled direction for concept $c_1$ given $c_2$ is:

$$\mathbf{w}_{c_1|c_2} = \mathbf{w}_{c_1} - \Pi_{c_2}(\mathbf{w}_{c_1}) \tag{28}$$

where $\Pi_{c_2}$ is the projection operator onto the subspace spanned by concept $c_2$:

$$\Pi_{c_2}(\mathbf{w}) = \mathbf{w}_{c_2} \frac{\mathbf{w}_{c_2}^T \mathbf{w}}{\|\mathbf{w}_{c_2}\|^2} \tag{29}$$

**Corollary A.1** (Bounds on Disentanglement Quality). For conditionally disentangled direction $\mathbf{w}_{c_1|c_2}$, the residual directional entanglement is bounded by:

$$\rho_{\mathrm{dir}}(\mathbf{w}_{c_1|c_2}, \mathbf{w}_{c_2}) \leq \frac{\|\mathbf{w}_{c_1} - \mathbf{w}_{c_1|c_2}\|}{\|\mathbf{w}_{c_1|c_2}\|} \tag{30}$$

**Proposition A.4** (Properties of Conditional CAVs). For conditionally disentangled CAV $\mathbf{w}_{c_1|c_2}$:

1. Orthogonality: $\mathbf{w}_{c_1|c_2}^T \mathbf{w}_{c_2} = 0$

2. Preservation: $\|\mathbf{w}_{c_1|c_2}\| \leq \|\mathbf{w}_{c_1}\|$

3. Interpretation: $\mathbf{w}_{c_1|c_2}$ represents concept $c_1$ after removing variance explained by $c_2$

*Proof Sketch.* Property (1) follows from the definition of orthogonal projection. Property (2) follows from the Pythagorean theorem. Property (3) is the operational interpretation of conditional disentanglement. $\square$

---

**Algorithm 1** Conditional CAV via Null-Space Projection

---

**Require:** Base CAV $\mathbf{w}_{c_1}$, conditioning CAVs $\mathbf{W}_{c_2} = [\mathbf{w}_{c_2}^{(1)}, \ldots, \mathbf{w}_{c_2}^{(m)}]$
**Ensure:** Conditionally disentangled CAV $\mathbf{w}_{c_1|c_2}$
  1: Compute projection matrix: $\mathbf{P} = \mathbf{W}_{c_2}(\mathbf{W}_{c_2}^T\mathbf{W}_{c_2})^{-1}\mathbf{W}_{c_2}^T$
  2: Compute projection: $\mathbf{w}_{\text{proj}} = \mathbf{P}\mathbf{w}_{c_1}$
  3: Compute residual: $\mathbf{w}_{c_1|c_2} = \mathbf{w}_{c_1} - \mathbf{w}_{\text{proj}}$
  4: Normalize: $\mathbf{w}_{c_1|c_2} = \frac{\mathbf{w}_{c_1|c_2}}{\|\mathbf{w}_{c_1|c_2}\|}$
  5: **return** $\mathbf{w}_{c_1|c_2}$

---

## B   Methodology Details

### B.1  Multi-Concept CAV Construction

In practice, MCAS subspaces are constructed by stacking individually trained CAVs and applying Gram-Schmidt orthogonalization, producing directions aligned with concept boundaries rather than maximum variance. The null-space projection operator supporting Algorithm 1 is:

$$\Pi_{\text{null}}(\mathbf{w}) = \mathbf{w} - \mathbf{W}_{c_2}(\mathbf{W}_{c_2}^T\mathbf{W}_{c_2})^{-1}\mathbf{W}_{c_2}^T\mathbf{w} \tag{31}$$

### B.2  Disentanglement Analysis

**Definition B.1** (Entanglement Metric)**.** The directional entanglement metric between concepts $c_1$ and $c_2$ (see Definition A.3) is defined as:

$$\rho_{\text{dir}}(\mathbf{w}_{c_1}, \mathbf{w}_{c_2}) = \frac{\mathbf{w}_{c_1}^T\mathbf{w}_{c_2}}{\|\mathbf{w}_{c_1}\|\|\mathbf{w}_{c_2}\|} \tag{32}$$

with $\rho_{\text{dir}} \in [-1, 1]$. Values near $\pm 1$ indicate strong directional entanglement (aligned directions), while values near 0 indicate directional independence. Note that this measures only angular alignment and does not capture overlap mass (see Definition A.4 and Lemma A.1).

The conditional variance measures residual entanglement:

$$\text{Var}(\mathbf{w}_{c_1}|\mathbf{w}_{c_2}) = \|\mathbf{w}_{c_1} - \Pi_{c_2}(\mathbf{w}_{c_1})\|^2 \tag{33}$$

The linear vs. nonlinear probe gap provides an entanglement indicator:

$$\Delta = L_{\text{nonlinear}} - L_{\text{linear}} \tag{34}$$

where $L_{\text{nonlinear}}$ and $L_{\text{linear}}$ are the losses of nonlinear and linear probes, respectively. A small gap indicates that entanglement is not merely a linear artifact.

**Proposition B.1** (Layer-wise Entanglement Accumulation)**.** For a transformer model with $L$ layers, let $\sigma_\ell$ denote the spectral norm of the weight matrix at layer $\ell$. The change in directional entanglement between adjacent layers is bounded by:

$$|\Delta\rho_{\text{dir}}^{(\ell)}| = |\rho_{\text{dir}}^{(\ell)} - \rho_{\text{dir}}^{(\ell-1)}| \leq 2\sigma_\ell \tag{35}$$

which follows from the Lipschitz continuity of cosine similarity under linear transformations. Accumulated entanglement is therefore bounded by $\rho_{\text{dir}}^{(L)} \leq \rho_{\text{dir}}^{(0)} + 2\sum_{\ell=1}^{L}\sigma_\ell$. Note that this bound applies to directional entanglement; measure entanglement may evolve differently across layers.

*Proof.* Let $\mathbf{w}_{c_1}^{(\ell)}, \mathbf{w}_{c_2}^{(\ell)}$ be concept directions at layer $\ell$. The proof proceeds under a local linearization of the transformer's layer transformation $g_\ell$: we treat $g_\ell$ as locally linear at the current activation, approximating it

by its Jacobian. This is a standard approximation in gradient-based analysis of transformers (e.g., Jacobian-based attribution methods) and is validated empirically in Table 7, which shows no bound violations across all tested models and layers. Under this approximation, $g_\ell$ has Jacobian bounded by spectral norm $\sigma_\ell$. Cosine similarity is Lipschitz with constant $\leq 2$ for unit vectors under linear perturbation: $|\cos\theta(\mathbf{u} + \boldsymbol{\delta}_1, \mathbf{v} + \boldsymbol{\delta}_2) - \cos\theta(\mathbf{u}, \mathbf{v})| \leq 2\max(\|\boldsymbol{\delta}_1\|, \|\boldsymbol{\delta}_2\|)$. Since the linearized layer transformation perturbs each direction by at most $\sigma_\ell$, the result follows. Deviation from linearity (due to attention's input-dependence and nonlinear activations) may tighten or loosen the bound in practice; Table 7 confirms the bound holds empirically across all evaluated configurations. $\qquad\square$

## B.3 Nonlinear Probes

**Definition B.2** (Nonlinear Probe). A nonlinear probe is a function $f_{\text{nonlinear}} : \mathbb{R}^d \to \mathbb{R}$ that maps activations to concept scores. Common formulations include:

$$f_{\text{nonlinear}}(\mathbf{e}) = g(\text{MLP}(\mathbf{e})) \quad \text{(MLP-based)} \tag{36}$$

$$f_{\text{nonlinear}}(\mathbf{e}) = K(\mathbf{e}, \cdot) \quad \text{(Kernel-based)} \tag{37}$$

where $g$ is an output function, MLP is a multi-layer perceptron, and $K$ is a kernel function.

**Proposition B.2** (Capacity Bounds of Nonlinear Probes). For an MLP-based nonlinear probe with $h$ hidden units and depth $d$, the VC dimension is bounded by $O(h^2 d)$, providing greater capacity than linear probes for separating entangled concepts.

*Proof.* This follows from standard VC dimension bounds for piecewise-linear networks. Bartlett et al. (2019) establish that the VC dimension of ReLU networks with $W$ total parameters is $O(WL\log W)$, where $L$ is the depth. For an MLP with $h$ hidden units per layer and $d$ layers, $W = O(h^2 d)$ and $L = d$, giving VC dimension $O(h^2 d^2 \log(h^2 d))$. Since linear probes have VC dimension equal to the input dimension $d_{\text{input}} + 1$, the nonlinear probe has strictly greater capacity when $h^2 d > d_{\text{input}}$, which holds for the architectures used in this work ($h = 128$, $d = 2$, $d_{\text{input}} = 768$). $\qquad\square$

**Definition B.3** (Separability Gap). The separability gap between linear and nonlinear probes is:

$$\Delta_{\text{sep}} = \text{Acc}_{\text{nonlinear}} - \text{Acc}_{\text{linear}} \tag{38}$$

where $\text{Acc}_{\text{nonlinear}}$ and $\text{Acc}_{\text{linear}}$ are the classification accuracies of nonlinear and linear probes, respectively.

**Corollary B.1** (Separability Gap as Entanglement Indicator). If $\Delta_{\text{sep}} \approx 0$ for a well-trained nonlinear probe, this indicates that directional entanglement is irreducible and cannot be resolved by increased model capacity. If $\Delta_{\text{sep}} > 0$, directional entanglement is reducible, suggesting that the observed entanglement is due to limited probe capacity rather than fundamental data correlations. This connects to Conjecture A.1: even when $\Delta_{\text{sep}} > 0$ (reducible directional entanglement), measure entanglement may persist.

**Corollary B.2** (Irreducible Entanglement Indicator). When $\Delta_{\text{sep}} \approx 0$ despite nonlinear probe capacity, directional entanglement is representationally intrinsic rather than a linear artifact. However, even when directional entanglement is reducible ($\Delta_{\text{sep}} > 0$), measure entanglement (activation distribution overlap) may persist due to fundamental data correlations, supporting Conjecture A.1's prediction for measure entanglement.

## B.4 Intervention Framework

**Definition B.4** (Concept-Aligned Perturbation). A Concept-Aligned Perturbation (CAP) for multi-concept intervention is:

$$\boldsymbol{\delta}^* = \operatorname{argmin}_{\boldsymbol{\delta}} \|\boldsymbol{\delta}\| \quad \text{subject to} \quad \mathbf{e} + \boldsymbol{\delta} \in \mathcal{M}_{\text{target}} \tag{39}$$

For multi-concept bias mitigation, the intervention is formulated as:

$$\mathbf{e}' = \mathbf{e} + \sum_{i=1}^{k} \alpha_i \mathbf{w}_i \tag{40}$$

---

**Algorithm 2** Multi-Concept Intervention

---

**Require:** Activation $\mathbf{e}$, MCAS $\mathbf{W} = [\mathbf{w}_1, \ldots, \mathbf{w}_k]$, target behavior $f_{\text{target}}$
**Ensure:** Perturbed activation $\mathbf{e}'$
 1: Initialize: $\boldsymbol{\alpha} = [0, \ldots, 0]^T$
 2: **while** $\|f(\mathbf{e}') - f_{\text{target}}\| > \epsilon$ **do**
 3:      Compute gradient: $\mathbf{g} = \nabla_{\boldsymbol{\alpha}} \|f(\mathbf{e} + \mathbf{W}\boldsymbol{\alpha}) - f_{\text{target}}\|^2$
 4:      Update: $\boldsymbol{\alpha} \leftarrow \boldsymbol{\alpha} - \eta\mathbf{g}$
 5:      Project: $\boldsymbol{\alpha} \leftarrow \text{clip}(\boldsymbol{\alpha}, \alpha_{\min}, \alpha_{\max})$
 6: **end while**
 7: Compute: $\mathbf{e}' = \mathbf{e} + \mathbf{W}\boldsymbol{\alpha}$
 8: **return** $\mathbf{e}'$

---

where $\alpha_i$ are intervention coefficients learned to achieve target behavior while minimizing perturbation magnitude.

**Definition B.5** (Intervention Fidelity)**.** Intervention Fidelity measures the success of intervention:

$$F = 1 - \frac{\|f(\mathbf{e}') - f_{\text{target}}\|}{\|f(\mathbf{e}) - f_{\text{target}}\|} \tag{41}$$

with $F \in [0, 1]$, where $F = 1$ indicates perfect intervention and $F = 0$ indicates no improvement.

**Proposition B.3** (Fidelity Bounds)**.** For linear interventions in MCAS $\mathbf{W}$ with bounded coefficients $|\alpha_i| \leq \alpha_{\max}$, the fidelity is bounded by:

$$F \geq 1 - \frac{\alpha_{\max}\|\mathbf{W}\|\|\nabla f(\mathbf{e})\|}{\|f(\mathbf{e}) - f_{\text{target}}\|} \tag{42}$$

**Proposition B.4** (Minimal Perturbation)**.** For a target CAR $\mathcal{M}_{\text{target}}$ and activation $\mathbf{e}$, the minimal perturbation achieving $\mathbf{e}' \in \mathcal{M}_{\text{target}}$ is:

$$\boldsymbol{\delta}^* = \text{argmin}_{\boldsymbol{\delta}:\mathbf{e}+\boldsymbol{\delta}\in\mathcal{M}_{\text{target}}} \|\boldsymbol{\delta}\| \tag{43}$$

For linear CARs, this reduces to orthogonal projection onto the target subspace.

*Proof Sketch.* For a linear CAR defined by $\{\mathbf{e} : \mathbf{W}^T\mathbf{e} \geq \boldsymbol{\tau}\}$, the minimal perturbation is the orthogonal projection onto the boundary, which can be computed via Lagrange multipliers. $\qquad\square$

## C  Threshold Sensitivity Analysis

Table 18 reports a systematic threshold sensitivity sweep. For each of 5 models $\times$ 5 seeds $\times$ 3 concept pairs, the threshold $\tau$ is varied from the 10th to the 90th percentile of CAV scores and $\rho_{\text{mass}}$ is recomputed. Three findings emerge. First, $\rho_{\text{mass}}$ varies monotonically with threshold as predicted by Lemma A.1: permissive thresholds ($\tau$@10th) yield $\rho_{\text{mass}} \approx 0.82$–$0.87$, while strict thresholds ($\tau$@90th) yield $\rho_{\text{mass}} \approx 0.02$–$0.08$. Second, $\rho_{\text{dir}}$ is invariant to threshold (by definition), confirming the metrics measure different quantities. Third, the cross-pair ranking is preserved at every threshold: toxicity-topic and gender-profession consistently show higher overlap than race-topic, regardless of the chosen threshold.

For the three-concept persistence analysis (gender $\times$ race $\times$ profession after orthogonalization), the pattern is similar: $\rho_{\text{mass}}^{\text{3-way}} = 0.74$ at $\tau$@10th, 0.43 at $\tau$@25th, 0.10 at $\tau$@50th, and $\approx 0$ at $\tau$@75th and above. The $\sim$40% figure reported in Experiment 6 corresponds to the median threshold; at more permissive thresholds the persistence is substantially higher. The key qualitative finding—that measure entanglement persists after complete directional orthogonalization—holds at all thresholds where the individual concept regions are non-degenerate ($\tau$@10th through $\tau$@50th).

Table 18: Threshold sensitivity analysis (R13). $\rho_{\mathrm{mass}}$ varies monotonically with threshold percentile while $\rho_{\mathrm{dir}}$ remains constant (by definition). Values averaged across 5 models × 5 seeds. The cross-pair ranking is preserved at every threshold: at $\tau$@50th, toxicity-topic (0.43) > gender-profession (0.46) > race-topic (0.30). Three-concept persistence after orthogonalization is nonzero at permissive thresholds ($\tau$@10th: 0.74, $\tau$@25th: 0.43) and approaches zero at strict thresholds ($\tau$@75th+), consistent with the theoretical prediction that measure entanglement depends on threshold choice (Lemma A.1).

| Metric | Threshold Percentile | | | | |
| | $\tau$@10th | $\tau$@25th | $\tau$@50th | $\tau$@75th | $\tau$@90th |
| --- | --- | --- | --- | --- | --- |
| *Pairwise $\rho_{mass}$ (cross-model average $\pm$ std)* | | | | | |
| Gender-profession | $0.856 \pm 0.011$ | $0.705 \pm 0.025$ | $0.464 \pm 0.010$ | $0.227 \pm 0.053$ | $0.081 \pm 0.063$ |
| Race-topic | $0.822 \pm 0.008$ | $0.582 \pm 0.017$ | $0.296 \pm 0.034$ | $0.098 \pm 0.020$ | $0.023 \pm 0.018$ |
| Toxicity-topic | $0.865 \pm 0.007$ | $0.688 \pm 0.012$ | $0.429 \pm 0.028$ | $0.190 \pm 0.014$ | $0.065 \pm 0.006$ |
| *$\rho_{dir}$ (threshold-invariant, shown for reference)* | | | | | |
| Gender-profession | | | $0.539 \pm 0.139$ | | |
| Race-topic | | | $0.031 \pm 0.030$ | | |
| Toxicity-topic | | | $0.783 \pm 0.044$ | | |
| *3-concept $\rho_{mass}^{3\text{-}way}$ after orthogonalization* | | | | | |
| Gender × Race × Prof | $0.738 \pm 0.034$ | $0.426 \pm 0.076$ | $0.102 \pm 0.082$ | $0.010 \pm 0.035$ | $0.000 \pm 0.000$ |

