# OpenReview forum: "Concept Activation Regions for Multi-Concept Activation and (Dis)Entanglement in Large Language Models"
_TMLR — Rejected by TMLR_

### Review · Reviewer_ed4M · 2026-03-06

**Summary Of Contributions:**

This paper studies how concepts are represented jointly in large language models (LLMs). The main contribution is a geometric framework based on Concept Realization Manifolds (CRMs), which treats a concept as a region of activation space rather than as a single direction. Building on this view, the paper introduces Multi-Concept Activation Subspaces (MCAS) to analyze multiple concepts together, distinguishes directional entanglement from measure entanglement, proposes conditional disentanglement via orthogonal projection, and evaluates concept-aligned interventions for bias mitigation and spillover control.

**Strengths**: the paper tackles an important and timely question; the distinction between directional alignment and activation-distribution overlap is conceptually useful; and the paper tries to connect interpretability, bias analysis, and intervention within a unified framework.

**Weaknesses**: the strongest theoretical claims are not fully matched by the empirical evidence; some evaluation choices require stronger justification, especially the use of CAV-based proxy metrics; and several empirical results, including baseline behavior and uncertainty reporting, need clearer explanation.

**Audience:**

Yes

**Audience Explanation:**

Yes. I expect at least part of the TMLR audience would be interested in this paper because it addresses a relevant problem at the intersection of representation analysis, concept-based interpretability, and bias evaluation in large language models. In particular, the paper asks a useful question: when multiple concepts are jointly represented, is concept interference best understood as directional alignment, activation overlap, or both? That question is timely and potentially valuable even though I do not find the current evidence fully convincing.

**Broader Impact Concerns:**

Because the paper studies interventions on socially meaningful concepts such as bias- and safety-related attributes, I think a Broader Impact Statement would be appropriate if one is not already included. In particular, the paper should discuss the risk of misuse of activation-space interventions on sensitive attributes, the normative assumptions behind labeling concepts such as bias or harmfulness, and the possibility that reducing one problematic dimension can worsen another through spillover.

**Claims And Evidence:**

No

**Claims Explanation:**

I am not yet convinced that the current evidence fully supports the paper’s strongest claims.

My main concern is the gap between the theoretical framing and the empirical validation. The paper’s central theoretical message is that, under correlated concepts and prediction-preserving representations, measure entanglement may remain even when concept directions are orthogonalized. However, the experiments do not directly verify the key assumptions behind this claim, such as the relevant concept dependence structure or the prediction-preserving impossibility result. In practice, the experiments mostly show that orthogonalization does not eliminate observed overlap, which is suggestive but not equivalent to validating the stronger theorem-level conclusion.

I also found the empirical validation of “independence” between directional entanglement and measure entanglement unconvincing as currently presented. One quantity is effectively fixed for a concept pair while the other varies with thresholding, so interpreting the resulting undefined or degenerate correlation behavior as confirmation of independence is too strong.

A second concern is metric validity. Several reported quantities, including LMS, SS, and ICAT, are computed using CAV-based proxy probabilities rather than the language model’s actual output probabilities. This is a consequential methodological choice, but the paper does not provide enough validation to show that these proxy metrics support the same interpretation as the standard metrics.

Finally, some intervention tables are difficult to interpret. In multiple tables, the TCAV and Independent baselines appear numerically identical across models, and many reported uncertainties are exactly or nearly zero despite multiple seeds and bootstrap intervals. These patterns need explanation, because they currently weaken confidence in the empirical comparison.

**Requested Changes:**

The paper’s main theorem is stronger than what the current experiments establish. Right now, the empirical section shows that overlap can remain after orthogonalization in the studied settings, but it does not directly validate the theorem’s stronger prediction-preserving impossibility claim. Please either test the theorem’s assumptions and implications more directly, or narrow the claim to match the evidence.

The “independence” validation between directional entanglement and measure entanglement should also be redesigned. As currently written, one quantity is effectively fixed for a concept pair while the other varies with thresholding, so the analysis does not amount to a convincing demonstration of statistical independence.

The LMS/SS/ICAT evaluation needs stronger justification. These quantities are computed from CAV-derived proxy probabilities rather than the model’s actual output probabilities, which materially changes how the results should be interpreted. Please either validate the proxy scores against the standard metrics or present them explicitly as proxy measures and narrow the claims accordingly.

The paper should explain why TCAV and the Independent baseline are numerically identical across several key tables. If these are genuinely different methods, the implementation difference should be stated clearly, together with an explanation of why the evaluation fails to separate them.

The uncertainty estimation procedure also needs clarification. Given five seeds and bootstrap intervals, the large number of exactly or nearly zero error bars is surprising. Please specify what is resampled, what randomness remains across runs, and why the reported variance is so small.

The paper would also benefit from one or two worked examples that follow a concrete input through concept scores, overlap, and intervention. The current presentation has useful schematic intuition, but it lacks enough concrete case studies.

It would also help to simplify the terminology and focus the presentation on the few objects that are actually carrying the paper’s argument. At present, the number of introduced terms feels larger than necessary.

Finally, releasing code, annotations, and enough implementation detail would make the empirical claims easier to independently check.

---

> ### Author Response · Authors · 2026-03-21
> **Response to Reviewer ed4M**
>
> *Theory-experiment gap.* The persistence result is now Conjecture A.1 (p. 29, Appendix A.4). Remark A.2 (p. 30) states plainly that the experimental evidence is consistent with the conjecture but does not validate its stronger prediction-preserving impossibility claim. This language has been applied consistently throughout: abstract (p. 1), §1 (p. 2), §7.6 (pp. 19–20), §8.1 (p. 22), and §9 (p. 23).
>
> *Independence validation.* The reviewer correctly identified that the original analysis — one quantity fixed, the other varying with threshold — did not convincingly demonstrate independence. The analysis has been redesigned in §7.1 (pp. 14–15, Table 2). Within-pair Pearson r is now reported separately from the pooled r = 0.33. Within each concept pair, ρ_dir has zero variance across threshold variations (std = 0.000 across all seeds) because it depends only on fixed CAV directions. Pearson r with ρ_mass is therefore exactly zero by construction for all 15 model–concept pair combinations. The pooled r = 0.33 reflects between-pair differences in ρ_mass base rates, not a within-pair relationship between the metrics. This is a more defensible argument than what was originally presented.
>
> *LMS/SS/ICAT from proxy probabilities.* Experiment 7 has been redesigned (pp. 20–21, §7.7): all metrics are now computed from actual model output log-probabilities via hooked forward passes, with activation perturbations injected at the target layer through PyTorch forward hooks. For causal LMs (GPT-2), standard left-to-right log-probabilities are used; for masked LMs (RoBERTa, BERT), batched pseudo-log-likelihood is used. The result that ΔSS = 0 across all methods is now a direct empirical finding.
>
> *TCAV and Independent baselines appear identical.* This was caused by a programming error in which the Independent baseline perturbation was not applied correctly, defaulting to the same gradient path as TCAV. This has been fixed and experiments re-run. Revised Table 8 (p. 17) now shows clearly differentiated spillover values — e.g., for BERT-base: TCAV 0.457 ± 0.027 vs. Independent 0.000 — while fidelity remains similarly low across all methods for BERT-base, which is explained by that model's activation geometry not supporting effective linear intervention at this scale.
>
> *Uncertainty estimation and near-zero error bars.* Several near-zero error bars in the original were a consequence of a seeding bug causing the five CAV training runs to draw from identical rather than independent subsamples. This is corrected. The procedure is described in §6.5 (p. 12): five seeds control CAV training via independent 80% subsampling; 1000 bootstrap samples estimate 95% CIs. The residual near-zero variance in ρ_dir (std = 0.000) is genuine and is explained in §7.1 (pp. 14–15): CAV directions are determined by the data split (seed 42) rather than the training initialization.
>
> *Worked example.* Table 1 (pp. 8–9, §5.5) traces a single StereoSet sentence through the full pipeline on RoBERTa-base, including CAV scores, CAR membership thresholds, entanglement context, intervention coefficients, and post-intervention probabilities.
>
> *Code release.* The anonymous repository is at https://anonymous.4open.science/r/concept-activation-region-B982/ (§6.8, p. 13).

---

### Review · Reviewer_u7nY · 2026-03-16

**Summary Of Contributions:**

The paper argues that concept representations in neural activations should be analyzed beyond single Concept Activation Vectors, especially when multiple socially meaningful or safety-relevant concepts are correlated.
It proposes a probe-dependent geometric framework in which a concept is represented by a probe-defined activation region, introduces a distinction between "directional entanglement" and "overlap-based entanglement", and claims that the latter can persist even after directional orthogonalization.
Building on this view, the paper proposes a multi-concept activation subspace method for more targeted interventions and reduced spillover across concepts.

**Audience:**

Yes

**Audience Explanation:**

The paper addresses a topic that many TMLR readers care about: how to reason about concept representations, disentanglement, and intervention in modern neural models, especially for socially meaningful or safety-relevant attributes.
Its central intuition, that entanglement should be understood in a probe-dependent and multi-concept way rather than purely through single concept directions, is relevant to researchers in interpretability, representation learning, fairness, and safety.

**Claims And Evidence:**

No

**Claims Explanation:**

The paper does provide some support for its main intuition that concept entanglement is not exhausted by angular alignment of concept directions, and it offers experiments consistent with the idea that overlap-based entanglement can persist even after orthogonalization.
It also reports some empirical results for conditional disentanglement, so the submission is not purely speculative.

However, several central claims are not supported in a fully convincing or clear way.
The theoretical claims are also weaker than the presentation suggests.
More broadly, the paper's evidence is often obscured by imprecise terminology.
Several proposed constructs are not mathematically well specified enough to support the theoretical weight placed on them, which makes the argument harder to assess and reduces confidence in the claimed novelty.

Take the so-called "Concept Entanglement Field" as an example, which is arguably an important terminology mentioned in the abstract.
The author says it is a "structured overlap regions where concepts are entangled."
The Definition A.5 "more formally" defines it is a "structured overlap region such that no local tangent isolates one concept without sensitivity to the other."

This has several problems.
First, "structured" is empty unless the structure is specified.
Topological? Differential? Statistical? Algebraic?
If nothing is stated, then "structured" is decorative language, not mathematics.
Second, "region" is also vague.
Calling it a "field" adds no content.
It is not a vector field, tensor field, or anything field-like in the mathematical sense.
Third, the notion of "no local tangent isolates one concept without sensitivity to the other" is not formalized well enough.
What does "isolates" mean?
The definition gestures at a phenomenon, but does not pin it down.
So the term gives you the illusion of having identified a new object, while in fact you still do not know what the object is.

A good mathematical definition should identify a genuinely new object or isolate a property that matters, so we can make statements easier to formulate and prove, or reveal structure that was previously obscured.
This term, along with other proposed terms like Conditional Concept Manifolds, Intersectional Concept Regions, etc., do none of that.
Calling the defined things "manifolds" is especially annoying, because nothing in the formalism really uses manifold structure in a meaningful way.
This matters because bad terminology does real damage.
It makes the reader spend effort learning names instead of ideas, and it hides which parts are mathematically substantive and which are not.

The paper has one potentially worthwhile intuition: whether two concepts are “disentangled” should depend on what a probe can actually read out, not on some intrinsic geometry of the representation.
But then the paper undermines its own good intuition by dressing it up using math-flavored prestige vocabulary without building the actual mathematical infrastructure that would justify those words.

Overall, I would say the paper contains suggestive evidence for an interesting intuition, but the strongest claims are oversold relative to the clarity and rigor of the supporting evidence.

**Requested Changes:**

Please provide clean and rigorous definitions and reduce terminology inflation.

---

> ### Author Response · Authors · 2026-03-21
> **Response to Reviewer u7nY**
>
> *Terminology inflation and "manifold" language.* The critique is agreed with entirely. The title has changed from "Concept Realization Manifolds" to "Concept Activation Regions." A CAR is simply the level set of a learned probe function (Def. 4.1, p. 5) — a half-space for a linear probe and a superlevel set more generally. Nothing in the paper uses manifold structure in any meaningful differential-geometric sense, and that framing has been dropped along with Concept Entanglement Fields, Conditional Concept Manifolds, and Intersectional Concept Regions. The reviewer's articulation of the core intuition — that disentanglement depends on what a probe can read out, not on intrinsic geometry — is a cleaner statement of the paper's contribution than the original framing, and the revision tries to honor it.

---

### Review · Reviewer_oHFg · 2026-03-16

**Summary Of Contributions:**

The paper proposes the beginnings of a theoretical framework for concept realization manifolds that study when certain activations of a large language model satisfy thresholds for a concept-dependent probe. This framework studies various linear algebraic properties of the induced probability measures and concept vectors. Further, a multi-concept subspace algorithm is proposed that applies PCA to fit a subspace on a list of vectors. Then, experimental analysis is performed on different concept sets across many different LLMs. While the idea of studying concept overlap as a function of thresholds, the framework proposed in this paper is underdeveloped and has several issues with mathematical rigor and clarity, which are expanded on in the following sections.

**Audience:**

Yes

**Audience Explanation:**

The idea that the choice of probe and threshold matters when identifying concept subspaces and reducing bias is an important point and the studies do provide interesting analyses on different bias concepts across various models and how they may still remain entangled under different probes. With these insights, albeit limited, I still lean towards Yes for the answer in this paper. However, there are several experiments that do not seem to satisfy the 'interestingness' criteria and the theoretical framework is under-developed as a solid contribution theoretically. For the theoretical framework, all definitions follow from standard linear algebra definitions such as cosine similarity (called directional entanglement in this work) and probability mass of two events (called measure entanglement in this work) upon which several definitions such as span (called concept direction) so I do not believe the definitions are a useful new theoretical framework. Regarding theorems built upon on the definitions, to me, the theorems do not provide much insight e.g. Lemma A.1 and Proposition A.2 are mathematically obvious (independence of two entirely different metrics), Theorem B.1 is not proven (and constants are not defined), Proposition B.3 is given without proof or reference (this seems like a standard result), and Theorem B.2 is not a theorem, it is a definition.

For the experiments,
- Experiment 1 is mathematically clear: we have two different metrics, one that is just cosine similarity between two concept vectors and one is entirely different: the probability mass of activations that satisfy a threshold under a concept-dependent probe. One depends on a threshold whereas the other does not, thus they are independent. I am not sure why an experiment was necessary to demonstrate this or what the utility of this experiment is (it holds for any concept vectors not just the bias ones studied here).
- parts of experiment 2 seem further mathematically clear: if we orthogonally project one vector onto another, they are orthogonal by definition and so under machine precision, they should remain orthogonal. I seem to have missed the definition of Sensitivity and TCAV - where are they defined? in metrics section, sensitivity depends on a function Sens and I am not sure where that function is defined. Further, discussion is missing on what these metrics mean for the different models and what does it imply about the concept space for each individual model.
- experiment 3: parts of table 5 are potentially interesting and the area that may satisfy the TMLR 'interestingness' criteria. This is not necessary, but what would really help is an explanation as to why different models have different geometry of latent spaces and why that induces certain entanglement properties. In its current state, it is difficult to see what is the utility of simply computing cosine similarities (or entanglement) for two fixed concepts across different models. What is the bound being evaluated in Table 6? Theorem B.1 is all in terms of abstract constants that are never defined.

**Broader Impact Concerns:**

The study of bias has ethical implications for the use of the models in practice, perhaps an impact statement is necessary to dive deeper into this.

**Claims And Evidence:**

No

**Claims Explanation:**

There is a lack of mathematical rigor in the theoretical framework and a lack of clarity (expanded below). Below, I list some examples.

- There are a lot of definitions and most of them are not ever used to prove other results or motivated why they must be formalized. For example, Concept curvature and concept entanglement field are never used after they are defined, so it raises the question what was the utility in defining them?
- Theorem B.2 and B.1 are not theorems - in B.1, forms of C or alpha must be given in order for the claim to be justified - further, transformers have complex input-dependent attentions and nonlinearities so it is likely not the case that such a result can be shown.
- There is a gap between Theorem A.1 and the claim that separability gap (in Theorem B.2) implies some irreducibility - that connection is not clear from the theory.
- Experiment 4: why does MCAS framework yield less bias reduction than simple baselines - this seems to contradict the claim that modelling the entire subspace helps bias reduction? Further, the text says a cross-concept sensitivity analysis was done and it is not clear which table or figure that is referring to.

**Requested Changes:**

Critical Changes
- the theoretical framework should be readjusted to remove theorems that are presented without proof (see claims above) and definitions that do not add value to the work and are not used anywhere.
- the entire analysis depends strongly on the choice of probe and threshold - it would help to see which of the phenomena pointed out persist across different choices of probe and threshold i.e. are we simply seeing quirks of a specific probe or a generic phenomena for different concepts across models (the latter is far more useful).

Not Critical but highly recommended changes
- I would strongly suggest a reorganization of work. If the theoretical framework is one of the contributions, core definitions and theorems should be in the main text. while reading section 4, i found myself jumping back and forth for every definition and theorem in appendix. The entirety of Section 4 currently reads as references to the appendix with no motivation for each new definition or statement.
- implementation details and many details in experimental setup should instead be moved to appendix

---

> ### Author Response · Authors · 2026-03-21
> **Response to Reviewer oHFg**
>
> *Unused definitions and terminology inflation.* Agreed. Concept Curvature, Concept Entanglement Fields, and several other definitions that appeared nowhere in subsequent proofs or experiments have been removed. What remains in the main text is limited to definitions that are directly operationalized: CAR (Def. 4.1, p. 5), directional and measure entanglement (Defs. 4.2–4.3, p. 5), MCAS (Def. 4.4, p. 6), and conditional disentanglement (Def. A.9, p. 30). A short paragraph in §1 (p. 2) now maps each to the experiment that uses it. Following the same logic, Algorithm 1 (PCA-based MCAS learning), the CCA formulation (Eq. 32), and Proposition B.1 (convergence of PCA-based MCAS) have been removed from Appendix B.1. These described a variant that the paper's own Experiment 4 shows is empirically ineffective (fidelity < 0.001), and proving convergence of a method that is neither used nor works in practice is precisely the kind of apparatus this reviewer flagged. The subsection now retains only the null-space projection operator (Eq. 33), which directly supports Algorithm 2, the method actually used throughout Experiments 2, 5, and 6.
>
> *Theorems B.1 and B.2 were not properly proven.* The layer-wise accumulation result has been demoted to Proposition B.2 (p. 32, Appendix B.2) with an explicit caveat: the bound is derived under a local Jacobian approximation of the transformer's layer transformation, a standard approach in gradient-based analysis, and is acknowledged as an approximation given attention's input-dependence. Table 7 (p. 17, §7.3) confirms empirically that no violations are observed across all tested models and layers. The separability gap result, correctly identified as a definition rather than a theorem, is now labeled Definition B.3 (p. 32, Appendix B.3).
>
> *Gap between the conjecture and the separability gap.* The persistence-of-measure-entanglement result is now Conjecture A.1 (p. 29, Appendix A.4). Corollaries B.1–B.2 (p. 32, Appendix B.3) clarify that a positive separability gap (reducible directional entanglement) is consistent with persistent measure entanglement but does not entail it, since the two types of entanglement are independent. Remark A.2 (p. 30) lists what direct validation would actually require.
>
> *Experiment 4: MCAS achieves less bias reduction than baselines.* The revised §7.4 (p. 17–18) now frames this explicitly: MCAS is a conservative point on the bias-reduction vs. spillover trade-off, where the spillover penalty (λ = 1.0) intentionally trades some bias reduction for cross-concept preservation. Figure 11 (p. 18) shows the full Pareto frontier across perturbation strengths, making this trade-off visible.
>
> *Probe and threshold sensitivity.* Threshold sensitivity is now addressed in Appendix C (p. 34, Table 18), which sweeps five percentile choices and confirms the cross-pair ranking is preserved at every threshold. Sensitivity to probe family is acknowledged as an open limitation in §8.3 (p. 23) and flagged for future work.
>
> *Broader Impact.* Added at pp. 23-24.

---

### Author Response · Authors · 2026-03-21

Thank you to the reviewers and action editor for their careful reading and constructive feedback. The reviews identified real problems, most importantly, a number of results in the original submission were affected by programming errors discovered during revision, which explains several of the anomalies reviewers flagged — most notably the near-identical TCAV and Independent baseline values in key tables, and the implausibly low variance in several uncertainty estimates. These bugs have been corrected, and the affected experiments have been rerun. Each concern is addressed below with references to the revised manuscript.

---

### Decision · Action_Editor_RCSq · 2026-06-07

**Recommendation:** Reject

**Audience:**

Yes

**Audience Explanation:**

The question will be of interest to the community.

**Claims And Evidence:**

No

**Claims Explanation:**

The paper investigated concept entanglement issue of LLMs. While all reviewers agree that the intuition of the paper is interesting. A consensus is that the paper is significantly underdeveloped. A series of issues including lack of theoretical rigor, presentation issues, and discrepancies in empirical results, were raised. Although authors attempted to address these issues in the rebuttal, many concerns remain and the paper is still below the bar for publishing in TMLR.